biomechanics/physiology

gait stability, external lateral stabilization, pelvis rotation, arm swing, step length, energy cost

**Author for correspondence:**
Mohammadreza Mahaki
e-mail: mahaki.mr@gmail.com; m.mahaki@vu.nl

# How does external lateral stabilization constrain normal gait, apart from improving medio-lateral gait stability?

Mohammadreza Mahaki[1,2], Trienke IJmker[2], Han Houdijk[2,3] and Sjoerd Matthijs Bruijn[2,4]

[1]Department of Sport Biomechanics, Faculty of Physical Education and Sport Sciences, Kharazmi University, Tehran, Iran
[2]Department of Human Movement Sciences, Faculty of Behavioural and Movement Sciences, Vrije Universiteit Amsterdam, Amsterdam Movement Sciences, van der Boechorststraat 9, Amsterdam, NL-1081 BT, The Netherlands
[3]Center for Human Movement Sciences, University Medical Centre Groningen, University Groningen, The Netherlands
[4]Orthopaedic Biomechanics Laboratory, Fujian Medical University, Quanzhou, Fujian, People's Republic of China

MM, 0000-0003-1439-2394; TI, 0000-0002-2363-6082; HH, 0000-0002-7069-1973; SMB, 0000-0003-0290-2131

Background: The effect of external lateral stabilization on medio-lateral gait stability has been investigated previously. However, existing lateral stabilization devices not only constrain lateral motions but also transverse and frontal pelvis rotations. This study aimed to investigate the effect of external lateral stabilization with and without constrained transverse pelvis rotation on mechanical and metabolic gait features. Methods: We undertook two experiments with 11 and 10 young adult subjects, respectively. Kinematic, kinetic and breath-by-breath oxygen consumption data were recorded during three walking conditions (normal walking (Normal), lateral stabilization with (Free) and without transverse pelvis rotation (Restricted)) and at three speeds (0.83, 1.25 and 1.66 m s$^{-1}$) for each condition. In the second experiment, we reduced the weight of the frame, and allowed for longer habituation time to the stabilized conditions. Results: External lateral stabilization significantly reduced the amplitudes of the transverse and frontal pelvis rotations, in addition to medio-lateral, anterior–posterior, and vertical pelvis displacements, transverse thorax rotation, arm swing, step length and step width. The amplitudes of free vertical moment, anterior–posterior drift over a trial, and energy cost were not significantly influenced by external lateral stabilization. The removal of pelvic rotation restrictions by our experimental set-ups resulted in normal

frontal pelvis rotation in Experiment 1 and significantly higher transverse pelvis rotation in Experiment 2, although transverse pelvis rotation still remained significantly less than in the Normal condition. Step length increased with the increased transverse pelvis rotation. Conclusion: Existing lateral stabilization set-ups not only constrain medio-lateral motions (i.e. medio-lateral pelvis displacement) but also constrain other movements such as transverse and frontal pelvis rotations, which leads to several other gait changes such as reduced transverse thorax rotation, and arm swing. Our new set-ups allowed for normal frontal pelvis rotation and more transverse pelvis rotation (yet less than normal). However, this did not result in more normal thorax rotation and arm swing. Hence, to provide medio-lateral support without constraining other gait variables, more elaborate set-ups are needed.

## 1. Introduction

Gait stability is achieved by interactions between the base of support and body centre of mass in the face of perturbations [1,2]. Both passive dynamics and active control are thought to be used by central nervous system to control gait stability [1,2]. Different stabilizing strategies such as ankle [3–5], foot placement [1,5–13], hip [9,14,15] and push-off [14,16] strategies are used to control gait stability. To better understand the control of gait stability, external lateral stabilization devices have been used to manipulate medio-lateral stability [6,17–22]. It has been reported that external lateral stabilization reduces medio-lateral centre of mass displacement [18] leading to lower need to control medio-lateral stability through the medio-lateral foot placement [6,17,18]. Some studies also reported that external lateral stabilization reduces metabolic energy cost of walking [17–19].

Existing lateral stabilization devices not only constrain medio-lateral motions, but also transverse and frontal pelvis rotations as well as (probably to a lesser degree) vertical pelvis displacement [17,21]. During normal walking, the leg has to produce forces and moments to compensate for rotations along the longitudinal axis, and the arms contribute to this by producing angular momentum in the opposite direction. Constraints of lateral external stabilization on transverse pelvis rotation may take away the need for the leg to produce such forces, and for the arms to counteract angular momentum produced by the swing leg [23]. In line with this, it has been reported that constrained pelvis rotation reduces the energy transfer from lower to upper extremities and subsequently reduces arm swing [24]. Transverse pelvis rotation also reduces vertical centre of mass displacement and contributes to step length [25], although only minimally [26]. Thus, apart from providing medio-lateral gait stability, lateral stabilization devices with constrained transverse pelvis rotation might limit the need to control angular momentum, reduces arm swing, and changes the vertical centre of mass displacement and step length. These potentially unwanted effects might change the interpretation of results reported by previous studies [17–19]. For example, the reduced energy cost in stabilized walking could be due to the aforementioned gait pattern constraints, rather than a reduced need to control medio-lateral gait stability [17–19].

The effect of constrained transverse pelvis rotation could be elucidated by adapting the standard lateral stabilization set-up to allow free pelvis rotations. Thus, in this study, we used two lateral stabilization conditions (with and without transverse pelvis rotation restriction) to determine how lateral stabilization set-ups constrain transverse pelvic rotation and concomitant gait features. We explored the direct effects of lateral stabilization on the amplitudes of the transverse and frontal pelvis rotations as well as the amplitudes of anterior–posterior and vertical pelvis displacements. Secondarily, we explored the indirect effects of the lateral stabilization on the amplitudes of transverse thorax rotation, arm swing, free vertical moment and step length. To investigate the effects of lateral stabilization on medio-lateral gait stability, the effect of lateral stabilization on medio-lateral pelvis displacement (as a direct effect of lateral stabilization) as well as step width and energy cost (as two indirect effects of lateral stabilization) were explored.[1]

## 2. Methods

Experiment 1 was performed to test the effect of external lateral stabilization with and without constrained transverse pelvic rotation on mechanical and metabolic gait features (figures 1 and 2*a*).

---

[1]Our initial research proposal can be found at https://osf.io/gkphs/.

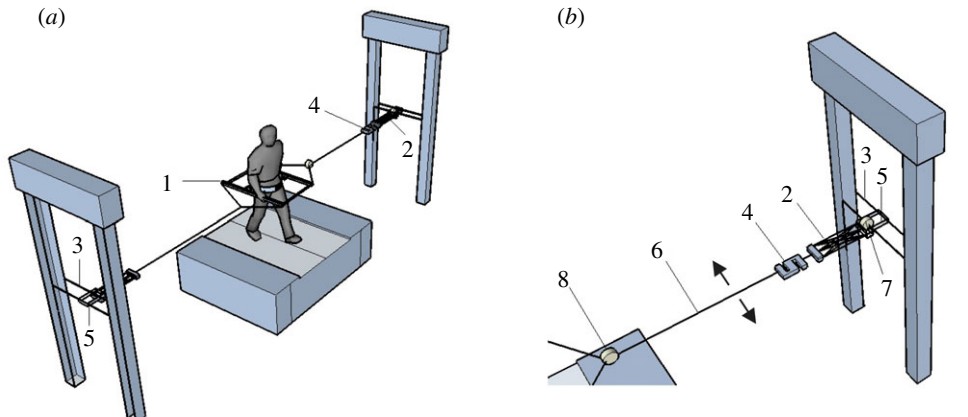

**Figure 1.** Schematic of experimental set-up used in Experiments 1 and 2. Inset (*b*) shows the stabilization in more detail. (1) Frame; (2) springs; (3) height-adjustable horizontal rail; (4) transducer (only used in additional experiments as reported in the electronic supplementary material); (5) ball-bearing trolley freely moving in anterior–posterior direction (arrows show the degree of freedom in anterior–posterior direction); (6) rope attached to frame; and (7 and 8) kinematics markers on the trolley and connection point between the springs and frame (only used in additional experiments as reported in the electronic supplementary material).

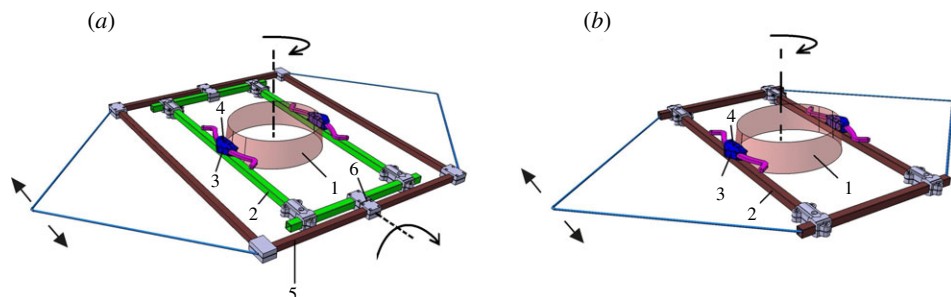

**Figure 2.** Schematic of the lateral stabilization frames used in Experiments 1 (*a*) and 2 (*b*). (1) Waist belt (we attached kinematic markers here); (2) inner frame which moves inside the outer frame to allow frontal pelvic rotation in Experiment 1; (3) slider between waist belt and inner frame to allow transverse pelvic rotation; (4) two screws resisting the sliders to work; (5) outer frame allowing normal arm swing; (6) the joint in which inner and outer frame were attached to each other and provided a free rotational degree of motion in the frontal plane. Arrows show the degrees of freedom around the related axes.

However, the results of Experiment 1 failed to reach the significant reduction of energy cost in the stabilized condition which was reported by some previous studies [17–19]. The potential effects of the frame weight on energy cost was considered as a potential confounding factor of Experiment 1. Additionally, the lack of habituation time to allow the participants for full familiarization with the set-up was considered as another reason for our inability to reduce energy cost in stabilized condition. Having the same aim and taking these potential confounding factors (weight of frame and habituation time) into account, we performed Experiment 2 to supplement Experiment 1. Moreover, additional experiments were performed to provide a detailed description of the characteristics of our set-up and the forces acting in it, which can be found in the electronic supplementary material.

## 2.1. Experiment 1

### 2.1.1. Participants

After signing the informed consent, a sample of 11 subjects (five men and six women, age: $27.5 \pm 2.4$ years, mass: $62.6 \pm 8.7$ kg, and height: $1.72 \pm 0.13$ m (mean ± s.d.)) participated in this study, which was approved by the local ethics committee.

### 2.1.2. Experimental set-up

A frame (weight = 4.5 kg) was used for the lateral stabilization. The frame was attached through a belt around the waist (figure 2*a*) and connected to bilateral springs of 1260 N m$^{-1}$, which were connected

to a rail placed at the height of the pelvis of the participant. This allowed free movements of the frame in the anterior–posterior direction. The frame included an inner and an outer frame, which were attached to each other and provided a free rotational degree of motion in the frontal plane between pelvis and frame. The distance between the inner and outer frames allowed normal arm swing during walking and participants were able to swing the arms through the full range of motion. To restrict/allow transverse pelvis rotation, two horizontal sliders between waist belt and inner frame were used. Two screws were embedded on each horizontal slider. In one condition, the screws were fastened and the pelvis was restricted from rotating in the transverse plane. In another condition, the screws were loosened and participants could rotate their pelvis with minimal friction between the waist belt and horizontal sliders on the inner frame.[2]

### 2.1.3. Experimental protocol

Participants performed three conditions (normal walking, entirely without wearing the frame and without being attached to the lateral stabilization set-up (Normal), lateral stabilization with transverse pelvis rotation restriction (Restricted), lateral stabilization without transverse pelvis rotation restriction (Free)). To generate the different levels of transverse pelvis rotation, all conditions were executed at three speeds (0.83, 1.25 and 1.66 m s$^{-1}$). Measuring at multiple speeds also has the added advantage of greater generalizability and increased statistical power. Participants were able to freely move their pelvis in frontal plane because of the joint between inner and outer frames. Thus, the frontal pelvis rotation was not restricted in both stabilized conditions (i.e. Free and Restricted). Each trial lasted 5 min and trials were separated by a resting period of approximately 2 min. Conditions were randomized following a two-step randomization procedure: for each participant, first, the conditions were randomized and then speeds were randomized. No familiarization was provided and participants were instructed to walk normally during stabilized walking.

## 2.2. Experiment 2

### 2.2.1. Participants

For Experiment 2, a sample of 10 subjects (six men and four women, age: 28.2 ± 3.7 years, mass: 70.3 ± 6.7 kg, and height: 1.77 ± 0.07 m (mean ± s.d.)) participated. Two of these participants also took part in Experiment 1. Experiment 2 was executed under the same ethical approval as Experiment 1.

### 2.2.2. Experimental set-up

The frame in Experiment 2 had no outer frame (figure 2$b$), and thus had a reduced weight (weight = 1.5 kg) compared to the frame used in Experiment 1. Two stiff ropes attached to the frame on either side, joined each other at 0.5 m from the frame, providing space for full range of motion that arms can swing. To this junction, two springs (1260 N m$^{-1}$) were attached. The other side of these springs was attached to the rail which allowed free movements in the anterior–posterior direction. This frame did not have a free degree of motion in frontal plane between pelvis and frame, thus restricted frontal pelvis rotation.

### 2.2.3. Experimental protocol

The protocol for Experiment 2 was the same as Experiment 1, except that participants were familiarized with walking in each condition for about 2 min. Data collection started 10 min after the end of the familiarization protocol. Participants were instructed to walk normally during stabilized walking.

## 2.3. Instrumentation

During Experiments 1 and 2, kinematic data were obtained from an Optotrak motion analysis system (Northern Digital Inc, Ontario, Canada), sampled at 100 samples s$^{-1}$. Clusters of three infrared markers were attached to the thorax (over the $T_6$ spinous process), the pelvis, the waist belt of the frame (figure 2$a$,$b$), the left and right arms (over the lateral and middle part of the humerus segment)

---

[2]The detailed description of friction between waist belt and horizontal sliders as well as between rail and ball-bearing trolley can be found in the electronic supplementary material.

and the heels. We also obtained kinetic data from the force plates embedded in the treadmill (ForceLink b.v., Culemborg, The Netherlands), sampled at 200 samples s$^{-1}$ in Experiment 2. During all experiments, participants wore a mask and breath-by-breath oxygen consumption was obtained using a pulmonary gas exchange system (Cosmed K4b$^2$, Cosmed, Italy).

## 2.4. Data processing

All our data and codes used to process the data for both experiments can be found at https://doi.org/10.5061/dryad.7pvmcvdrr.

The direct effects of lateral stabilization were explored by calculating the amplitudes of transverse and frontal pelvis rotations as well as the amplitudes of medio-lateral, anterior–posterior and vertical pelvis displacements. To explore the indirect effects of lateral stabilization, the amplitudes of transverse thorax rotation, arm swing, free vertical moment, step length, step width and energy cost were calculated.

Kinematic data from the Optotrak system were not filtered, but large jumps in the data were removed, and gaps of less than 10 samples were interpolated using a shape-preserving spline algorithm. The displacement of the markers attached to each segment were used to quantify segment displacement. Heel strike events were identified as the minimum in the vertical position of the heel marker, and identified heel strike events were visually inspected. Each gait cycle was defined from left heel strike to subsequent left heel strike. Orientation of each segment was determined by a local anatomical reference frame constructed using three segment markers [27]. Using Euler angles (zxy sequence), the time series of transverse and frontal pelvis rotations, and transverse thorax rotation were calculated from the segment orientation matrices. The time series of angular (i.e. transverse and frontal pelvis rotations, and transverse thorax rotation) and displacement (i.e. medio-lateral, anterior–posterior, vertical pelvis displacements and arm swing (i.e. anterior–posterior position of arm with respect to anterior–posterior position of thorax)) variables were time normalized to 0–100% for each gait cycle. The amplitudes of these variables were calculated as the differences between maximum and minimum angles/displacements per gait cycle and then median of amplitudes over gait cycles for each trial. Moreover, to explore whether lateral stabilization constrains the anterior–posterior drift of participants over trial, we calculated the amplitude of anterior–posterior drift as the differences between maximum and minimum of anterior–posterior pelvis displacements over a walking trial. For arm swing, we calculated the average over arms since non-significant differences were found between left and right arms.

Ground reaction force data, collected in Experiment 2, were filtered with a 5 Hz cut-off frequency (second order, bidirectional Butterworth digital filter). The time series of ground reaction forces were time normalized to 0–100% for each gait cycle. The vertical ground reaction moment (also referred to as 'free vertical moment', e.g. Li et al. [28]) was calculated as the moment around a vertical axis caused by the interaction between the feet and the floor. The amplitude of free vertical moment was calculated as the differences between maximum and minimum vertical ground reaction moments per gait cycle and then the median of amplitudes over gait cycle for each trial. Step length and step width were defined as the median of the distances between both feet in anterior–posterior and medio-lateral foot directions at heel strike, respectively. For the step length, we calculated the average over legs since non-significant differences were found between left and right step lengths. Step width was calculated likewise as the average over left and right steps.

To evaluate the effect of lateral stabilization on energy cost, net energy cost was calculated. Oxygen uptake ($\dot{V}O_2$; ml min$^{-1}$), carbon dioxide production ($\dot{V}CO_2$; ml min$^{-1}$) and respiratory exchange ratio (RER) were determined with a pulmonary gas exchange system. The metabolic rate reached a plateau within the 5 min trial, as was confirmed through visual inspection. To be consistent with literature and to ensure that our outcome was not influenced by any potential noise, respiratory gases were averaged over the last 3 min of each trial. We calculated gross metabolic rate ($E_{gross}$; J min$^{-1}$) using the following equation [29]:

$$E_{gross} = ((4.940 \cdot RER + 16.04) \cdot \dot{V}O_2)/ \text{ body mass (kg)}.$$

Resting metabolic rate, determined during seated position for 5 min prior to the trials, was subtracted from gross metabolic rate to calculate net metabolic rate during walking. To calculate net energy cost (J kg$^{-1}$ m$^{-1}$), net metabolic rate was divided by walking speed (m min$^{-1}$).

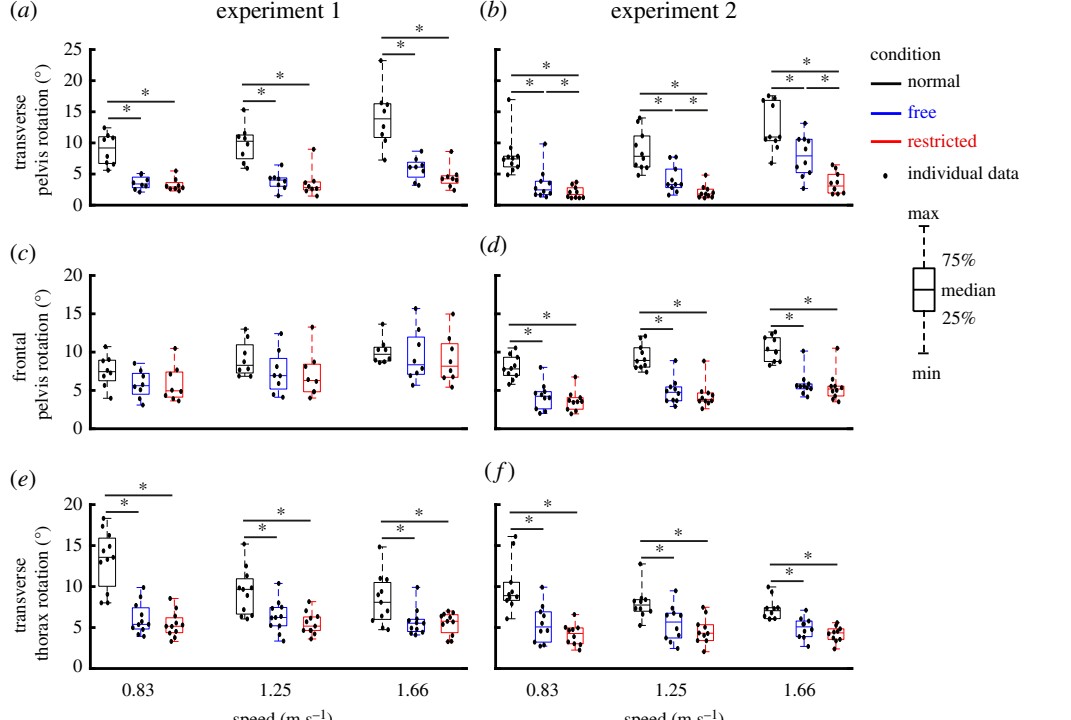

**Figure 3.** Group average transverse pelvis rotation (*a,b*) and frontal pelvis rotation (*c,d*) as well as transverse thorax rotation (*e,f*) (median ± 25th percentile) at each walking speed in the Normal (black), stabilized conditions without transverse pelvis rotation restriction (blue) and with transverse pelvis rotation restriction (red) in Experiments 1 and 2. * denotes significant differences between conditions (based on the results of Bonferroni correction of Condition effect). Individual data are also plotted as dots.

## 2.5. Statistical analysis

The Shapiro–Wilk tests confirmed the normal distribution of data for most of the trials ($p > 0.05$). Thus, repeated measures ANOVA with Condition (Normal, Free and Restricted) and Speed (0.83, 1.25 and 1.66 m s$^{-1}$) as factors was used to test for the effects of external lateral stabilization with and without transverse pelvic rotation and walking speed on all direct and indirect outcomes. When a significant main effect of Condition was found, a *post hoc* test with Bonferroni correction was applied to determine the differences between conditions. We assessed the Condition × Speed interaction, to test for the differences in the effects of external lateral stabilization with and without transverse pelvis rotation on aforementioned parameters between different walking speeds. For F-test, $\eta_p^2$ was reported as effect size. Level of significance for all statistical analyses was set at $p < 0.05$. We did not perform statistical comparisons between Experiments 1 and 2 since the participants, study set-ups and protocols were different for the two experiments.

## 3. Results

### 3.1. Experiment 1

Direct effects of lateral stabilization were found in the amplitudes of transverse pelvis rotation (figure 3*a*), medio-lateral, anterior–posterior and vertical pelvis displacements (figure 4*a,c,e*), which were significantly influenced by Condition (table 1). *Post hoc* analyses showed that in Free and Restricted conditions, the amplitudes of transverse pelvis rotation, medio-lateral, anterior–posterior and vertical pelvis displacements were significantly reduced when compared to the Normal condition (table 1). However, the differences of influenced gait variables between Free and Restricted conditions were not significant (table 1). The amplitude of anterior–posterior drift over a trial (figure 5*a*) was not significantly influenced by Condition (table 1).

Indirect effects of lateral stabilization were found in the amplitudes of transverse thorax rotation (figure 3*e*), arm swing (figure 6*a*), step length (figure 7*a*) and step width (figure 7*c*), which were

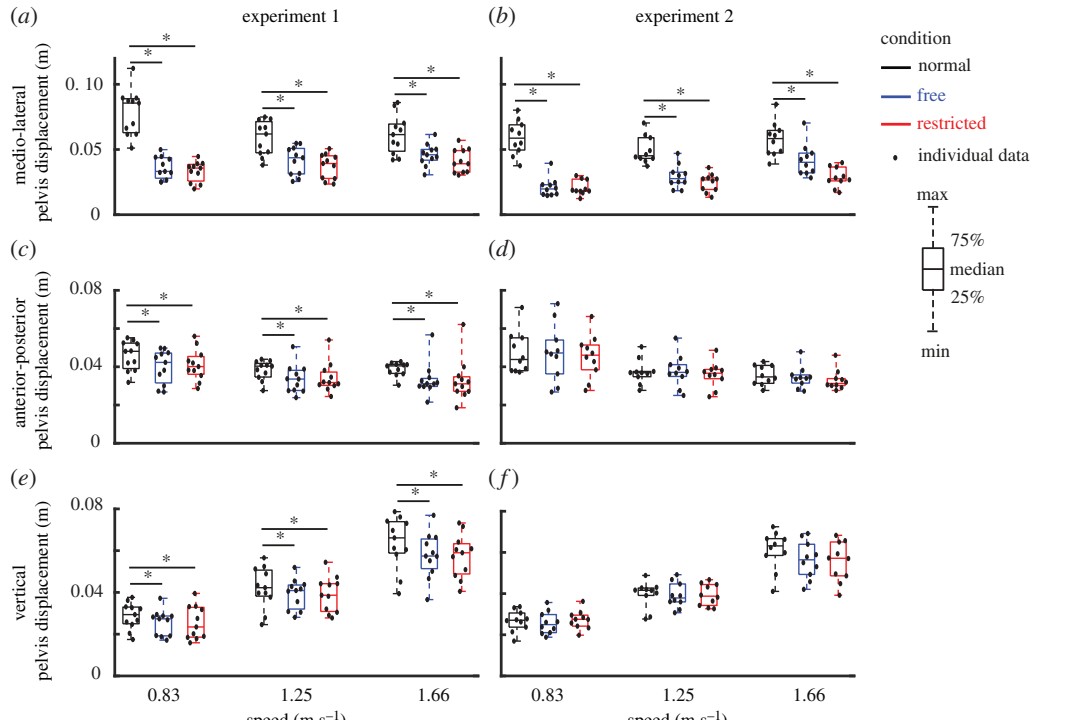

**Figure 4.** Group average pelvis displacements (median ± 25th percentile) at each walking speed in the Normal (black), stabilized conditions without transverse pelvis rotation restriction (blue) and with transverse pelvis rotation restriction (red) for medio-lateral (a,b) and anterior–posterior (c,d) as well as vertical directions (e,f) in Experiments 1 and 2. * denotes significant differences between conditions (based on the results of Bonferroni correction of Condition effect). Individual data are also plotted as dots.

significantly influenced by Condition. Energy cost (figure 8a) was not significantly influenced by Condition (table 1). *Post hoc* analyses showed that in Free and Restricted conditions, the influenced gait variables were significantly reduced when compared to the Normal condition (table 1). However, the differences of influenced gait variables between Free and Restricted conditions were not significant (table 1).

At faster walking speeds, the amplitudes of transverse and frontal pelvis rotations (figure 3a,c), anterior–posterior and vertical pelvis displacements (figure 4c,e), transverse thorax rotation (figure 3e), arm swing (figure 6a), step length (figure 7a), step width (figure 7c) and energy cost (figure 8a) increased (Speed effect and related *post hoc* analyses; table 1). The increases of transverse pelvis and thorax rotations (figure 3a,e), and arm swing (figure 6a) with increasing walking speed were shown to be more prominent in Normal condition when compared to the Free and Restricted conditions (Condition × Speed effects; table 1). No significant differences in transverse pelvis rotation and arm swing with increasing walking speed were found between Free and Restricted conditions (Condition × Speed effects; table 1).

### 3.2. Experiment 2

Direct effects of lateral stabilization were found in the amplitudes of transverse pelvis rotation (figure 3b), frontal pelvis rotation (figure 3d) and medio-lateral pelvis displacement (figure 4b), which were significantly influenced by Condition (table 2). *Post hoc* analyses showed that the amplitude of transverse pelvis rotation was significantly higher in Normal than in Free and in Free than in Restricted condition (figure 3b and table 2). In Free and Restricted conditions, frontal pelvis rotation and medio-lateral pelvis displacement were significantly reduced when compared to the Normal condition (table 2). The amplitudes of anterior–posterior and vertical pelvis displacements (figure 4d,f) were not significantly influenced by Condition (table 2). The amplitude of anterior–posterior drift over a trial (figure 5b) was not significantly influenced by Condition (table 1).

Indirect effects of lateral stabilization were found in the amplitudes of transverse thorax rotation (figure 3f), arm swing (figure 6b) and step width (figure 7d), which were significantly influenced by

**Table 1.** The direct and indirect effects of external lateral stabilization on gait features in Experiment 1. Bold indicates significant results.

| type of effects | variables | condition effect | | | speed effect | | | condition × speed effect | | | post hoc comparisons - condition | | | post hoc comparisons - speed | | |
|---|---|---|---|---|---|---|---|---|---|---|---|---|---|---|---|---|
| | | F(1, 2) | P | $\eta_p^2$ | F(1, 2) | P | $\eta_p^2$ | F(1, 4) | P | $\eta_p^2$ | normal-free | normal-restricted | free-restricted | 0.83–1.25 (m s⁻¹) | 0.83–1.66 (m s⁻¹) | 1.25–1.66 (m s⁻¹) |
| direct effects | transverse pelvis rotation (deg) | 36.23 | **<0.001** | 0.84 | 18.95 | **<0.001** | 0.73 | 4.79 | **0.005** | 0.41 | **<0.001** | **<0.001** | 1.00 | 1.00 | **<0.001** | **<0.001** |
| | frontal pelvis rotation (deg) | 3.06 | 0.08 | 0.30 | 16.51 | **<0.001** | 0.70 | 1.69 | 0.18 | 0.19 | 0.22 | 0.11 | 1.00 | 0.06 | **<0.001** | **0.02** |
| | medio-lateral pelvis displacement (m) | 37.60 | **<0.001** | 0.79 | 3.15 | 0.07 | 0.24 | 18.02 | **<0.001** | 0.64 | **<0.001** | **<0.001** | 0.87 | 0.16 | 1.00 | 0.10 |
| | anterior–posterior pelvis displacement (m) | 5.68 | **0.01** | 0.36 | 25.91 | **<0.001** | 0.72 | 0.25 | 0.91 | 0.02 | **0.02** | **0.04** | 1.00 | **<0.001** | **<0.001** | 1.00 |
| | vertical pelvis displacement (m) | 6.37 | **0.007** | 0.39 | 220.74 | **<0.001** | 0.96 | 2.00 | 0.11 | 0.17 | **0.02** | **0.01** | 1.00 | **<0.001** | **<0.001** | **<0.001** |
| | anterior–posterior drift over trial (m) | 2.53 | 0.11 | 0.20 | 0.87 | 0.44 | 0.08 | 0.47 | 0.75 | 0.05 | 0.11 | 0.69 | 0.98 | 1.00 | 0.92 | 0.72 |
| indirect effects | transverse thorax rotation (deg) | 31.56 | **<0.001** | 0.76 | 5.46 | **0.01** | 0.35 | 21.39 | **<0.001** | 0.68 | **<0.001** | **<0.001** | 1.00 | 0.12 | **0.01** | **0.94** |
| | arm swing (m) | 31.68 | **<0.001** | 0.76 | 48.95 | **<0.001** | 0.83 | 3.13 | **0.03** | 0.24 | **<0.001** | **<0.001** | 1.00 | **<0.001** | **<0.001** | **0.01** |
| | step length (m) | 3.57 | **0.04** | 0.26 | 862.09 | **<0.001** | 0.99 | 1.67 | 0.18 | 0.14 | **0.04** | 0.30 | 1.00 | **<0.001** | **<0.001** | **<0.001** |
| | step width (m) | 37.60 | **<0.001** | 0.79 | 24.20 | **<0.001** | 0.72 | 17.60 | **<0.001** | 0.64 | **<0.001** | **<0.001** | 1.00 | 0.57 | **<0.001** | **<0.001** |
| | energy cost (J kg⁻¹ m⁻¹) | 1.02 | 0.38 | 0.10 | 11.83 | **<0.001** | 0.57 | 0.91 | 0.47 | 0.09 | 0.79 | 1.00 | 0.62 | 1.00 | **0.003** | **0.001** |

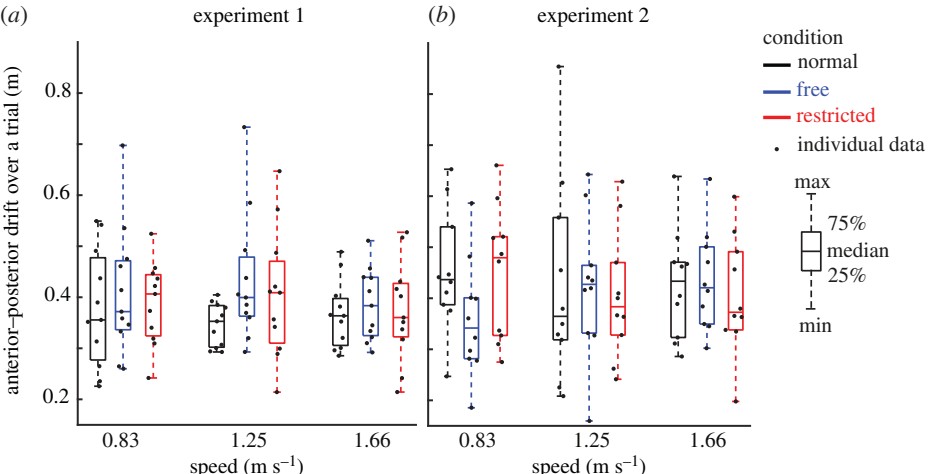

**Figure 5.** Anterior–posterior drift over a trial (median ± 25th percentile) at each walking speed in the Normal (black), stabilized conditions without transverse pelvis rotation restriction (blue) and with transverse pelvis rotation restriction (red) in Experiments 1 (*a*) and 2 (*b*). Individual data are also plotted as dots.

**Figure 6.** Group average arm swing (in Experiments 1 (*a*) and 2 (*b*)) and free vertical moment (in Experiment 2 (*c*)) (median ± 25th percentile) at each walking speed in the Normal (black), stabilized conditions without transverse pelvis rotation restriction (blue) and with transverse pelvis rotation restriction (red). * denotes significant differences between conditions (based on the results of Bonferroni correction of Condition effect). Individual data are also plotted as dots.

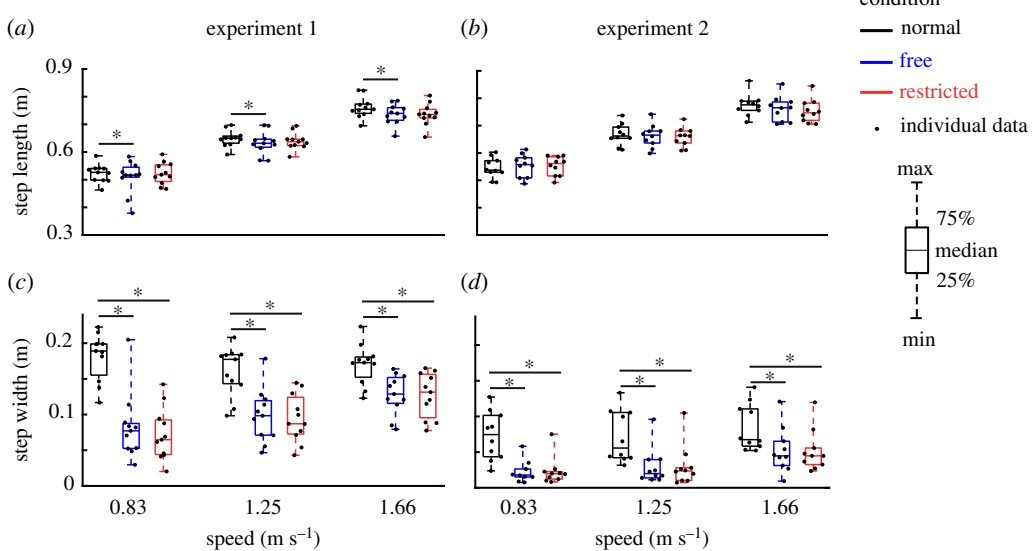

**Figure 7.** Group average step parameters (median ± 25th percentile) at each walking speed in the Normal (black), stabilized conditions without transverse pelvis rotation restriction (blue) and with transverse pelvis rotation restriction (red) for step length (*a,b*) and step width (*c,d*) in Experiments 1 and 2. * denotes significant differences between conditions (based on the results of Bonferroni correction of Condition effect). Individual data are also plotted as dots.

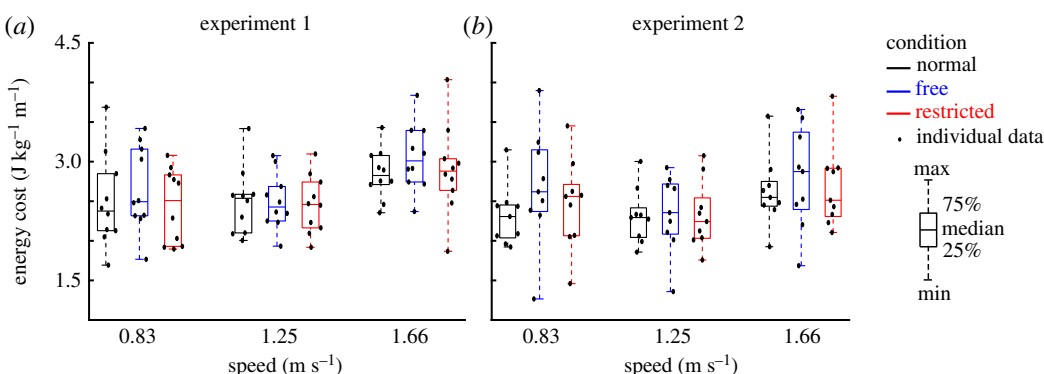

**Figure 8.** Group average energy cost (median ± 25th percentile) at each walking speed in the Normal (black), stabilized conditions without transverse pelvis rotation restriction (blue) and with transverse pelvis rotation restriction (red) in Experiments 1 (*a*) and 2 (*b*). Individual data are also plotted as dots.

Condition (table 2). Free vertical moment (figure 6*c*), step length (figure 7*b*) and energy cost (figure 8*b*) were not significantly influenced by Condition (table 2). *Post hoc* analyses showed that the differences of transverse thorax rotation, arm swing, and step width between Free and Restricted conditions were not significant (table 2). In both Free and Restricted conditions, transverse thorax rotation, arm swing and step width were significantly reduced when compared to the Normal condition (table 2).

At faster walking speeds, the amplitudes of transverse and frontal pelvis rotations (figure 3*b,d*), medio-lateral, anterior–posterior and vertical pelvis displacements (figure 4*b,d,f*), transverse thorax rotation (figure 3*f*), arm swing (figure 6*b*), free vertical moment (figure 6*c*), step length (figure 7*b*), step width (figure 7*d*) and energy cost (figure 8*b*) increased (Speed effect and related *post hoc* analyses; table 2). The increase of transverse pelvis rotation with increasing of walking speed was more pronounced in the Normal condition than in the Free condition as well as in Free condition than in Restricted condition (Condition × Speed effect; figure 3*b* and table 2). The increases of medio-lateral pelvis displacement, transverse thorax rotation, arm swing and step width with walking speed were more pronounced in the Normal condition when compared to the Free and Restricted conditions (Condition × Speed effect; table 2).

**Table 2.** The direct and indirect effects of external lateral stabilization on gait features in Experiment 2. Bold indicates significant results.

| type of effects | variables | condition effect | | | speed effect | | | condition × speed | | | post hoc comparisons - condition | | | post hoc comparisons - speed | | |
|---|---|---|---|---|---|---|---|---|---|---|---|---|---|---|---|---|
| | | $F_{(1, 2)}$ | P | $\eta_p^2$ | $F_{(1, 2)}$ | P | $\eta_p^2$ | $F_{(1, 4)}$ | P | $\eta_p^2$ | normal-free | normal-restricted | free-restricted | 0.83–1.25 (m s⁻¹) | 0.83–1.66 (m s⁻¹) | 1.25–1.66 (m s⁻¹) |
| direct effects | transverse pelvis rotation (deg) | 31.67 | **<0.001** | 0.78 | 22.96 | **<0.001** | 0.72 | 5.37 | **0.002** | 0.37 | **<0.001** | **<0.001** | **0.03** | 1.00 | **<0.001** | **<0.001** |
| | frontal pelvis rotation (deg) | 53.08 | **<0.001** | 0.86 | 27.46 | **<0.001** | 0.75 | 2.06 | 0.106 | 0.19 | **<0.001** | **<0.001** | 1.00 | **0.008** | **<0.001** | **0.003** |
| | medio-lateral pelvis displacement (m) | 54.60 | **<0.001** | 0.86 | 18.00 | **<0.001** | 0.66 | 11.50 | **<0.001** | 0.56 | **<0.001** | **<0.001** | 0.20 | 1.00 | **<0.001** | **<0.001** |
| | anterior–posterior pelvis displacement (m) | 1.43 | 0.27 | 0.14 | 18.22 | **<0.001** | 0.67 | 0.18 | 0.95 | 0.02 | 1.00 | 0.44 | 0.54 | **<0.001** | **<0.001** | 0.73 |
| | vertical pelvis displacement (m) | 2.44 | 0.12 | 0.21 | 224.43 | **<0.001** | 0.96 | 7.09 | **<0.001** | 0.44 | 0.16 | 0.31 | 1.00 | **<0.001** | **<0.001** | **<0.001** |
| | anterior–posterior drift over trial (m) | 0.58 | 0.57 | 0.06 | 0.007 | 0.99 | 0.001 | 1.69 | 0.17 | 0.16 | 0.88 | 1.00 | 1.00 | 1.00 | 1.00 | 1.00 |
| indirect effects | transverse thorax rotation (deg) | 30.89 | **<0.001** | 0.77 | 4.17 | **0.03** | 0.32 | 5.43 | **0.002** | 0.38 | **<0.001** | **<0.001** | 0.27 | 0.57 | **0.03** | 0.44 |
| | arm swing (m) | 43.97 | **<0.001** | 0.83 | 29.19 | **<0.001** | 0.76 | 8.09 | **<0.001** | 0.47 | **<0.001** | **<0.001** | 0.16 | **<0.001** | **<0.001** | **0.01** |
| | free vertical moment [N m] | 0.48 | 0.63 | 0.06 | 192.90 | **<0.001** | 0.96 | 1.69 | 0.18 | 0.18 | 1.00 | 1.00 | 1.00 | **<0.001** | **<0.001** | **<0.001** |
| | step length (m) | 2.32 | 0.127 | 0.21 | 514.83 | **<0.001** | 0.98 | 5.18 | **0.002** | 0.37 | 1.00 | 0.14 | 0.69 | **<0.001** | **<0.001** | **<0.001** |
| | step width (m) | 22.71 | **<0.001** | 0.72 | 16.18 | **<0.001** | 0.64 | 3.40 | **0.02** | 0.27 | **<0.001** | **<0.001** | 1.00 | 1.00 | **<0.001** | **0.001** |
| | energy cost (J kg⁻¹ m⁻¹) | 1.456 | 0.263 | 0.15 | 8.15 | **0.004** | 0.51 | 2.72 | **0.047** | 0.25 | 0.33 | 1.00 | 0.95 | 0.31 | 0.11 | **0.003** |

# 4. Discussion

We used external lateral stabilization with and without transverse pelvis rotation to investigate how existing lateral stabilization set-ups constrain gait apart from providing medio-lateral gait stability. Our results showed that external lateral stabilization with and without transverse pelvis rotation not only constrains medio-lateral motions (i.e. the amplitudes of medio-lateral pelvis displacement and step width), but also constrains transverse pelvis rotation, which coincided with a reduced transverse thorax rotation, and reduced arm swing which could be considered an indirect consequence of the restricted transverse pelvis rotation. The removal of pelvic rotation constraints in our experimental set-up led to increased frontal pelvis rotation in Experiment 1 and also increased transverse pelvis rotation in Experiment 2. Our results demonstrated that amplitudes of anterior–posterior drift over a trial, free vertical moment and energy cost were not influenced by lateral stabilization.

Using two transverse sliders between waist belt and inner frame, our results showed that transverse pelvis rotation was less reduced by lateral stabilization in the Free condition than in the Restricted condition (i.e. the set-up used by previous studies) in Experiment 2. Our results also showed that frontal plane pelvis rotation was not reduced by lateral stabilization with the frame used in Experiment 1. The frame used in Experiment 1 included an inner and an outer frame, which were attached to each other and provided a free rotational degree of motion in the frontal plane between pelvis and frame in all stabilized conditions; however, the frame used in Experiment 2 did not have an outer frame and it was more similar to the frame used by previous studies [21,22]. Therefore, in contrast to the set-ups used by previous studies in which frontal and transverse pelvis rotations were restricted, our set-up that allowed free frontal and transverse pelvis rotations (Experiment 1) resulted in normal frontal plane pelvis rotation. While our set-up that only allowed free transverse pelvis rotation (Experiment 2) resulted in more normal (i.e. greater) transverse plane pelvis rotation.

The reduced frontal and transverse pelvic rotations might confound the interpretation of previously reported results. For instance, our new set-ups resulted in normal frontal (Experiment 1) and more transverse pelvis (Experiment 2) rotations. The reduced frontal and transverse pelvis rotations in previous studies and partly in our set-up, need to be considered as the results of the physical constraints of apparatus and cannot be attributed to a strategy to control gait stability. Moreover, it has been reported that transverse pelvis rotation, as one of the gait determinants [30], has some mechanical and metabolic benefits as it reduces vertical centre of mass displacement [25,30] and increases step length [25]. There is no clear consensus, however, on whether reduced vertical centre of mass displacement induces a metabolic cost reduction [25,30] or not [31,32]. Moreover, the effect of transverse pelvis rotation on step length is still disputed throughout the literature [25,26]. Assuming that transverse pelvis rotation reduces vertical centre of mass displacement and metabolic cost and that transverse pelvis rotation increases step length, restricted transverse pelvis rotation induced by lateral stabilization might offset these benefits. In line with this, the restricted step length in our Experiment 1 could be due to the restricted transverse pelvis rotation, and unaffected step length in our Experiment 2 could be considered as the mechanical benefits of increased transverse pelvis rotation.

The reduced vertical pelvis displacement could be due to downward/upward forces induced by lateral stabilization which might also confound the interpretation of reported results. If bilateral springs provide downward forces, subjects might need some extra energy cost to compensate for them, which consequently might offset the energy cost savings of external lateral stabilization. Conversely, the potential upward forces induced by springs might decrease the body weight which could increase energy cost savings of lateral stabilization. To avoid downward/upward forces, one possibility is to use longer rope to attach the springs to the frame. An alternative possibility may be to attach the springs to rails that can also move in the vertical direction, although this may also induce unwanted oscillations.

Consistent with Matsubara *et al*. [33], our results confirmed that external lateral stabilization did not constrain the amplitude of anterior–posterior drift over a trial because bilateral springs were connected to a horizontal trolley that could move freely and in phase with the displacement of the centre of mass in anterior–posterior direction. However, springs fixed in anterior–posterior direction used by previous studies [17–19,34] may provide unwanted assistance in the anterior direction (see electronic supplementary material, figure S1) and subsequently decreases the net propulsive force generating by the subjects (cf. [35]). It has been reported that humans try to decrease levels of muscle activation during walking in force fields which has also been called 'slacking' [36,37]. Thus, in previous studies with fixed springs [17–19,34], subjects may have benefited from slacking in the stabilized condition if

they discovered that walking is less energetically costly at the back of the treadmill than at other spots. Here, unwanted anterior component forces pull the subject forward, potentially leading to lower levels of muscle activation and reduced propulsive force. Although some studies tried to minimize these potential unwanted effects by using long ropes to connect the springs to the subjects (i.e. 8.5 and 14.5 m [17,19], respectively), other studies with shorter ropes (i.e. 3.0 and 4.0 m [18,34], respectively) might have increased these effects and subsequently some of the energy cost savings of external lateral stabilization found in these studies may have been due to 'slacking'. Our proposed system with trolleys avoids such effects from happening.

Although subjects were allowed to perform arm swing and they were instructed to walk as naturally as possible, our results showed reduced arm swing in stabilized conditions. It has been reported that the restriction of transverse pelvis rotation reduces the energy transfer between the lower and upper limb segments and subsequently deceases the amplitude of arm swing [24]. The reduced arm swing might be due to the remaining restricted transverse pelvis rotation in stabilized conditions since our new set-up did not lead to fully normal transverse pelvis rotation. However, arm swing was less restricted in the present study than those studies in which subjects were not allowed to swing their arms [17,18] and they had to walk with their arms crossed (to avoid contacting the external stabilizer). Our results also showed that free vertical moment was not influenced by lateral stabilization. This indicates that the rate of angular momentum was not different between conditions [23]. Previous studies which provided lateral stabilization with completely constrained pelvis rotation and arm swing [18,38] might have limited the need to control of angular momentum as hypothesized by [23]. It can be concluded that our set-up and the set-ups used by previous studies not only constrain medio-lateral motions (i.e. medio-lateral pelvis displacement and step width), but also transverse and frontal pelvis rotation, thereby indirectly leading to decreases in transverse thorax rotation and arm swing. This should be considered when interpreting the results of these studies.

Consistent with previous studies [6,17,18,22], our results confirmed that external lateral stabilization provided medio-lateral gait stability, represented by reductions of medio-lateral pelvis displacement and step width in stabilized condition. However, in contrast to some of the previous studies [17,19], we did not find a decrease in energy cost during stabilized walking. This appears to contradict the notion that medio-lateral stabilization is an active process which entails energy cost. However, there may be several explanations for this. Firstly, springs that are fixed in anterior–posterior direction as used by previous studies [17,19] might provide unwanted forces and assistance in the anterior direction and might have increased some of the energy cost savings of external lateral stabilization. Secondly, it has been reported that with arm swing the effect of lateral stabilization on energy cost was slightly lower (a significant 3–4% reduction), compared to walking without arm swing (a significant 6–7% reduction) [19]. Therefore, arm swing restriction induced by set-up used by previous studies [17,18] might also have increased some of the energy cost savings of lateral stabilization. Thirdly, providing more than 30 min habituation time, Ortega *et al*. [19] reported a significant 3–4% reduction of energy cost in stabilized condition. However, we did not provide any habituation time in Experiment 1 which was considered as a potential confounding factor. Therefore, we performed Experiment 2 to allow the participants to become more familiarized with walking in stabilized conditions. Compared to Ortega *et al*. [19], a shorter habituation time (i.e. between 3 and 10 min) was used in our Experiment 2 and in the previous studies with the reported non-significant effect of lateral stabilization on energy cost [18,22]. Therefore, a too short habituation time to allow the participants for full familiarization might be responsible for our inability to reduce energy cost in stabilized conditions. Lastly, although the potential effects of the frame weight on energy cost were minimized by removing outer frame in Experiment 2, the frames used by IJmker *et al*. [21,22] and our two experiments may have added additional energy cost, and subsequently these factors might offset some of the reduced energy cost induced by external lateral stabilization. To draw a conclusion about these observed effects of lateral stabilization on energy cost, a meta-analysis on the available data could provide valuable insights.

# 5. Conclusion

External lateral stabilization set-ups do not only constrain the amplitudes of the medio-lateral motions but also the amplitudes of the transverse and frontal pelvis rotations, leading to reduced transverse thorax rotation as well as arm swing. Although the removal of constraints by our experimental set-up led to normal frontal and increased transverse pelvis rotations, other concomitant gait variables such as transverse thorax rotation and arm swing still remained restricted in stabilized conditions.

Therefore, future studies are recommended to take these limitations of lateral stabilization set-ups into consideration and to improve their set-ups.

Research ethics. This study had been approved by the local ethics committee of the Faculty of Behavioural and Movement Sciences of the Vrije Universiteit, Amsterdam (VCWE-2017-154).

Data accessibility. All our data and codes used to process the data for both experiments can be found at https://doi.org/10.5061/dryad.7pvmcvdrr [39].

Authors' contributions. M.M. conceived and designed the experiments, performed the experiments, analysed the data, prepared figures and/or tables, approved the final draft. T.I. and H.H. conceived and designed the experiments, analysed the data, contributed reagents/materials/analysis tools, authored or reviewed drafts of the paper, approved the final draft. S.M.B. conceived and designed the experiments, analysed the data, contributed reagents/materials/analysis tools, prepared figures and/or tables, authored or reviewed drafts of the paper, approved the final draft.

Competing interests. We declare we have no competing interests.

Funding. S.M.B. was funded by a VIDI grant (no. 016.Vidi.178.014) from the Dutch Organization for Scientific Research (NWO). M.M. was also funded by a grant from Ministry of Science, Research and Technology of Iran.

Acknowledgements. We like to thank Leon Schutte and Hans de Koning for experimental help and the participants for their time.

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
