## [Peer Review File · Royal Society Open Science]

Review History

RSOS-200647.R0 (Original submission)

Review form: Reviewer 1

Is the manuscript scientifically sound in its present form?

Yes

Are the interpretations and conclusions justified by the results?

Yes

Is the language acceptable?

Yes

Do you have any ethical concerns with this paper?

No

Have you any concerns about statistical analyses in this paper?

No

Recommendation?

Major revision is needed (please make suggestions in comments)

Comments to the Author(s)

This paper demonstrates that external lateral stabilization constrains normal gait, providing effects aside from improving mediolateral gait stability. Furthermore, this paper proposes and tests two novel designs for a new apparatus for mediolateral stabilization to negate the unwanted effects seen in previous experiments. Specifically, previous mediolateral stabilization apparatuses directly led to a reduction in transverse and frontal pelvis rotations and vertical pelvis displacement. The first proposed design had additional degrees of freedom to allow for frontal and transverse pelvis rotation, whereas the second design only had one additional degrees of freedom to allow for transverse pelvis rotation. However, the smaller frame of the second design was intended to unconstrain arm swing. The two designs were tested separately with two different subject groups in three conditions (normal, free, and restricted) at three different speeds. The first design did not result in any significant differences in frontal and transverse rotation of the pelvis between the free and restricted conditions. However, both conditions significantly reduced pelvis rotation and mediolateral displacements compared to the normal condition. Indirectly, the mediolateral stabilization led to reduced transverse thorax rotation, arm swing, and step width. The results of the second design were similar, however, the free condition resulted in a more transverse pelvis rotation than the restricted condition but not as much as seen in the normal condition. The paper concluded that more elaborate set-ups are required to provide mediolateral stability without constraining other aspects of gait.

This paper thoroughly described and analyzed the changes that were elicited by the mediolateral stabilization. The two designs proposed in this study were novel yet very intuitive that would appear to provide a possible solution to the issues with transverse and frontal pelvis rotation. Additionally, this paper takes the opportunity to explore the results of past publications, where the results observed may have been previously misinterpreted or incomplete. However, there are some major concerns outlined below that should be addressed, which made the results and implications of their work more difficult to follow.

Major Comments:

The first concern is about Experiment 1 and 2. When the authors introduced the two different experiments, they state that Experiment 2 considered the confounding factors of Experiment 1. However, the differences between the two are not clearly described in methods, and the discussion should include a larger explanation of the specific confounding factors. Without this understanding and based on the "confounding factors" statement, it almost seems Experiment 1 should be part of Appendix, as there is a fair amount of results to track between the two experiments. Based on the two designs, I gathered that the second experiment aimed to allow for arm swing which could have possibly encouraged larger amounts of transverse pelvic rotation. Assuming that is true, discussion of the decoupled result of increased transverse pelvic rotation without a significant difference in arm swing should be provided. If this was not the motivation of Experiment 2, the authors should provide what confounding factors they hoped to account for and if they were successful in doing so. In addition, although both the experiments aimed to provide additional degrees of freedom, they were not successful in allowing for the motion. Can the authors comment on the possible reasons that led to the motion still being constrained?

The second concern is about expanding on the implications of the results. In the discussion, the authors suggested that other studies must take into account the additional constraints elicited from the lateral stabilization. This is an interesting statement and is one of the main reasons for this study. However, the implications of the additional constraints are not fully addressed and need to be further explored. Specifically, how does the direct or indirect effects of lateral

stabilization could lead to (or influence) the energetic results of the experiments in this study and previous studies?

As the aim of the device is to provide lateral stabilization, can the authors comments on whether or not they believe that the additional effects observed may be a result of the stabilization rather than the physical constraints of apparatus? For example, it has previously been thought that the reduction in step-width is a result of the lateral stabilization as the stepping strategy is supplemented. Could the same be true for the frontal pelvis rotation, supplementing the trunk strategy for maintaining stability? This may also provide an explanation as to why when the additional degree of freedom was included in Experiment 1, additional pelvis rotation was not observed.

The third concern is that, based on the discussion and the results, there is some lack of clarity on some of the results of the study. This is partially due to the lack of clarity between Experiment 1 and Experiment 2, but I believe further clarification is required to make sure the results are clearly communicated.

Page 14, Line 19-22: Please specify which experiment this result was obtained for. Additionally, an increase transverse and frontal plane pelvis rotation was reported, but according to Table 1 and 2, only the second experiment led to a significant increase in the transverse pelvic rotation of comparing the free and restricted conditions. There does not appear to be any significant difference in the frontal pelvis rotation between the restricted and free conditions. I am unsure if my understanding of results is incorrect or if this statement is incorrect.

Page 15, Line 33 to 38: Was there any statistical analysis between the two different experiments? It does appear that the amplitude is higher looking at Fig. 3 C and D, but it would be helpful to specify whether the statement is qualitative or quantitative.

Slight modifications to the figures and tables would greatly improve that readability of the paper.

Figure 2: The degrees of freedom of the two different apparatuses are a little unclear in the images. Adding arrows to show the motion of the degrees of freedom about the one or two axes of rotation would enhance the readability of the image.

Table 1 & 2: A caption for these two tables would make the manuscript easier to read so the reader can keep track of which table corresponds to which set of experiments.

Figure 3: The caption should specify that it is angular limb kinematics.

Figure 4: The caption should be more specific to the figure saying group average pelvis displacements.

Figures 3-7: Adding asterisks to show a significant difference between conditions of the same speed would make the results of the analysis much clearer.

There are some clarifications needed for the methods section as well.

Could the authors comment on what they hoped to accomplish by conducting the experiment at multiple speeds?

Page 10, Line 3: Were the displacements normalized at all to either the subject height or leg length?

I believe that certain subjects brought up in the introduction of the paper need to be revisited or further supported.

Page 5, Line 6-8: The definition of gait stability provided is not necessarily true as gait does not have one perfect measure (the manuscript's reference 2 explores this in depth); where many measures are not just about the base of support and centre of mass. It might be better to qualify the gait stability as achieved with interactions between the base of support and centre of mass rather than defining it strictly.

Minor Comment: Page 5, Line 27: Lateral does not need to be capitalized.

Review form: Reviewer 2

Is the manuscript scientifically sound in its present form?

Yes

Are the interpretations and conclusions justified by the results?

No

Is the language acceptable?

Yes

Do you have any ethical concerns with this paper?

No

Have you any concerns about statistical analyses in this paper?

Yes

Recommendation?

Major revision is needed (please make suggestions in comments)

Comments to the Author(s)

The objective of the current study was to determine whether previously employed methods to confer external lateral stabilization also reduce transverse plane rotation of the pelvis, which could confound the interpretation of previous studies (namely on medial-lateral COM displacement and metabolic cost).

It is important to note that the the cited methods of external lateral stabilization are primarily used to examine lateral balance control (i.e., medial-lateral COM displacement and foot placement control) and associated the associated metabolic cost, the indirect measures described in the current study. The premise is that by providing external lateral stabilization, study participants won't have to control medial-lateral step to step balance, allowing an assessment of the metabolic cost associated with medial-lateral step-to-step balance control.

Novel methods are introduced to both restrict and allow free transverse plane rotation of the pelvis. There are a number of interesting results, yet the main motivation for the study, limiting transverse plane pelvic rotation could explain the decrease in ML COM displacement, need for foot placement control, and the associated metabolic cost, are not addressed in the discussion or conclusion. A number of other concerns require attention as well.

1. My primary concern is that the main motivation for the study as presented in the introduction (i.e., traditional methods of external lateral stabilization may also limit transverse plane pelvic motion, which may in turn confound the interpretation of previous studies using these methods to investigate key aspects of medial-lateral balance control (COM displacement, foot placement control, and metabolic cost) is not addressed at all. In fact, the results suggest the opposite, that transverse plane pelvic rotation has little or no effect on these three metrics. Thus, it would appear that the existing interpretation of those studies stands. Despite being a major component to the motivation for the study, this point is not acknowledged nor discussed within the manuscript. Based on the introduction as written, this would seem to be the main result. As written, the last paragraph in the discussion ignores this key point, and goes on to suggest that in fact we should have concern when interpreting the results of those prior studies. The data presented do not seem to support those concerns and hence conclusions. Please revise.

2. It seems as though there is perhaps a missed opportunity to discuss what if any metabolic benefit exists to transverse pelvic rotation. This could be brought up in the context of the original determinants of gait (Kerrigan et al.)

a. Kerrigan DC, Riley PO, Lelas JL, Della Croce U. Quantification of pelvic rotation as a determinant of gait. *Arch Phys Med Rehabil* 2001;82:217-20.

b. Della Croce U, Riley PO, Lelas JL, Kerrigan DC. A refined view of the determinants of gait. *Gait and Posture* 2001;14(2):79-84.

3. Much of the discussion consists of rehashing the results. The discussion could be much more focused, thereby shortening the paper overall.

4. Please confirm the normality, or lack therefore, in the data (i.e., suitability of parametric tests). It is also odd to report medians but then use parametric tests. Please explain.

Minor

1. There are probably better references that could be used when citing ankle, foot placement, and hip balance control strategies.

2. Plots in Figure 3 could probably be increased in size to improve visibility.

Review form: Reviewer 3 (Hendrik Reimann)

Is the manuscript scientifically sound in its present form?

No

Are the interpretations and conclusions justified by the results?

No

Is the language acceptable?

Yes

Do you have any ethical concerns with this paper?

No

Have you any concerns about statistical analyses in this paper?

No

Recommendation?

Major revision is needed (please make suggestions in comments)

Comments to the Author(s)

The authors used a device, in two variants, to provide elastic support to participants during walking. Participants wore a waist belt attached to a frame that was connected to lateral supports by springs mounted on horizontal sliders. Two experiments tested two different frames that differed in allowing or constraining rotation in the frontal plane. Both frames allowed to either lock or release a constraint on rotation in the transverse plane. The main goal was to investigate the effects of lateral stabilization on mechanical and metabolic gait features, and whether the constraint in the transverse plane makes a difference. The results do show an effect of stabilization on some gait parameters, but not others that other studies found affected, most notably energy cost. Releasing the transverse constraint increased the rotation in that direction, but not up to the level during normal walking.

This study is one incremental step in a larger body of work investigating lateral mechanical stabilization during walking. One problem with lateral stabilization is to design a device that constrains some degrees of freedom in the desired fashion, while leaving others unconstrained. This study makes some interesting contributions to this field, but suffers from presenting these in a way that I found somewhat hard to follow, which might be a symptom of an underlying lack of clarity in the experimental design.

The main goal of this study is to compare the effect of lateral stabilization with and without a constraint on pelvis rotations in the transverse plane, implemented by locking or releasing lateral sliders that allow the connection point between the pelvis and the stabilization frame on either side to move in the anterior-posterior direction, labelled “free” and “restricted” walking. This is compared with a third condition of “normal” walking, without any stabilization. The authors performed two experiments, where Experiment 2 differed from Experiment 1 by (a) adding a two-minute familiarization period for each condition and (b) using a mechanically simpler stabilization device that removed the outer frame, thus making the device lighter, but also constraining pelvis rotation in the frontal plane. I don’t really understand the rationale for two different experiments here. The authors state that “Experiment 2 supplemented experiment 1, as it considered several potential confounding factors in the design and set-up of experiment 1” – but these confounding factors are not listed or discussed in detail. The main difference is the removal of the outer frame in the device, thus constraining pelvis rotations in the frontal plane. This seems to *add* a potential confounding factor, rather than removing it. I can see how the absence of a familiarization period in Experiment 1 is a potential confounding factor, but on its own, I wouldn’t say that provides sufficient rationale for a whole new experiment. Could you please elaborate in what manner the addition of the frontal plane constraint removes a confounding factor, and what other confounding factors that Experiment 2 avoided?

Another factor is that in the device used here, the connection between the body-worn frame and the external anchoring frame was not fixed, but mounted on trolleys that were free to move in the anterior-posterior direction. This, the authors argue, removes a back door for using the lateral springs to generate forward-pulling forces that decrease the metabolic cost of walking. This is a very interesting point, because this would imply that the results from other experiments showing that lateral stabilization reduced metabolic cost might be spurious results of the forces in the anterior-posterior direction implicitly generated by the mechanism. This point is only made in the discussion, maybe because the authors also only realized this after seeing their results? If this was something the authors were aware of before conducting the experiment, I would suggest adding this to the list of research questions they attempted to answer. Otherwise, I suggest that you clarify that this result was surprising and the explanation was generated post-hoc. In the latter case, I would also recommend bringing this up a bit later in the Discussion section: currently it is

the first phenomenon that is discussed in-depth, after the general first paragraph, suggesting that this is the main result, which is at odds with it not being mentioned as a research question earlier.

The authors made all data and analysis scripts available, which is great. Looking over what's in this cloud drive, however, I noticed some weird issues. One issue is that some of the variables have very large jumps, as seen in software/Plots, for example the Right Arm Swing in Subject 1, Trial 1 or Subject 14, Trial 5. This might be a problem with calculating angles from the rigid body orientation given by three markers on the cluster, since the jumps seem to be roughly around 90deg. Similar jumps occur in ML Pelvis Displacement, though, e.g. Subject 1, Trial 6; Subject 5, Trial 9; Subject 9, Trial 4. Another issue is gaps in the data, where some of the trajectories will just disappear for some of the gait cycle, e.g. in Right Arm Swing in Subject 10, Trial 9 or Transverse Pelvis Rotation Subject 1, Trial 4, where around 60% of the gait cycle *all* data is missing, similarly for Subject 6, Trial 5 around 0-20%. I did not go through the analysis code in detail, so it is possible that these are just intermediate results, before such issues have been weeded out by the authors, although the readme.docx seems to suggest that this is not the case. If this is the case and these artifacts are still part of the data as analyzed in the manuscript, then I suggest that the authors go back to the data processing stage and take a close and careful look at where they come from and how to avoid them. In some cases, removing a small number of problematic gait cycles might be sufficient, but in other cases, all data seems to be missing for part of the gait cycle, and I don't know of a good way to deal with this.

Minor issues and questions

p.7, l.32 "the pelvis was restricted from rotating in the transverse plane" – what exactly was the mechanism here to release or lock this constraint? Figure 2 seems to imply that some slider can be locked. Please add some detail here.

p.7, l.45 "normal walking" – did normal walking consist of walking while wearing the device but without lateral springs attached, or of walking entirely without the device? Please add this information to the text. If the latter, did you confirm that the normal arm swing was not impeded by the frame at all? The hands can reach quite far ahead of the body in normal arm swing, especially at fast walking speeds, and from Figures 1 and 2 it seems that arm swing might be impeded.

p.9, l.8 "Clusters of three infrared markers were attached to ... the left and right arms" – please specify where exactly on the arms the clusters were placed.

p.9, l.29/50 Please specify which convention you used for axes of rotation in calculating Euler angles.

p.9, l.46 – how did you identify the heel strike events?

p.10, l.10f: anterior-posterior pelvis displacement – at this point it seems odd that the ap-displacement is defined differently than the other displacements/angles. Is this because the ap-displacement is relevant in terms of the implicit forces in this direction that were *not* applied by your device due to the trolley connection? If so, it would help to explain that at this point, and maybe give this variable a different name to more clearly differentiate it from the other means.

p.10, l.35 "distances between ... foot placements" – what exactly was the location of the foot placement here?

p.10, l.37 "for the step length, we calculated the average over legs, since nonsignificant differences were found between left and right step lengths" – what about step width? If it was not significant, did you also pool data? If it was significant, did you use it as a factor in the statistical analysis?

Most figures: The legend indicates that the whiskers of the box-and-whisker plots cover the whole range of the data, from "Min" up to "Max". However, many of the individual data points are outside of this range, so that cannot be correct. Please clarify. Also, it would help to add some horizontal jitter to the individual data points, so they are still distinguishable when multiple data points are close together.

Language

p.6, l.1 constrains → constraints

p.9, l.55 matrixes → matrices

multiple locations: capitalize "Experiment 1"

– Hendrik Reimann

Decision letter (RSOS-200647.R0)

Dear Dr Mahaki:

Manuscript ID RSOS-200647 entitled "How does external lateral stabilization constrain normal gait, apart from improving medio-lateral gait stability?" which you submitted to Royal Society Open Science, has been reviewed. The comments from reviewers are included at the bottom of this letter.

In view of the criticisms of the reviewers, the manuscript has been rejected in its current form. However, a new manuscript may be submitted which takes into consideration these comments.

Please note that resubmitting your manuscript does not guarantee eventual acceptance, and that your resubmission will be subject to peer review before a decision is made.

Your resubmitted manuscript should be submitted by 10-Jan-2021. If you are unable to submit by this date please contact the Editorial Office.

on behalf of Dr Manoj Srinivasan (Associate Editor) and Pietro Cicuta (Subject Editor)
openscience@royalsociety.org

Associate Editor Comments to Author (Dr Manoj Srinivasan):
Associate Editor: 1

Comments to the Author:

We look forward to a revised draft addressing the reviewer concerns. It looks like all the reviewer comments are addressable.

Associate Editor: 2

Comments to the Author:

(There are no comments.)

Reviewers' Comments to Author:

Reviewer: 1

Comments to the Author(s)

This paper demonstrates that external lateral stabilization constrains normal gait, providing effects aside from improving mediolateral gait stability. Furthermore, this paper proposes and tests two novel designs for a new apparatus for mediolateral stabilization to negate the unwanted effects seen in previous experiments. Specifically, previous mediolateral stabilization apparatuses directly led to a reduction in transverse and frontal pelvis rotations and vertical pelvis displacement. The first proposed design had additional degrees of freedom to allow for frontal and transverse pelvis rotation, whereas the second design only had one additional degrees of freedom to allow for transverse pelvis rotation. However, the smaller frame of the second design was intended to unconstrain arm swing. The two designs were tested separately with two different subject groups in three conditions (normal, free, and restricted) at three different speeds. The first design did not result in any significant differences in frontal and transverse rotation of the pelvis between the free and restricted conditions. However, both conditions significantly reduced pelvis rotation and mediolateral displacements compared to the normal condition. Indirectly, the mediolateral stabilization led to reduced transverse thorax rotation, arm swing, and step width. The results of the second design were similar, however, the free condition resulted in a more transverse pelvis rotation than the restricted condition but not as much as seen in the normal condition. The paper concluded that more elaborate set-ups are required to provide mediolateral stability without constraining other aspects of gait.

This paper thoroughly described and analyzed the changes that were elicited by the mediolateral stabilization. The two designs proposed in this study were novel yet very intuitive that would appear to provide a possible solution to the issues with transverse and frontal pelvis rotation. Additionally, this paper takes the opportunity to explore the results of past publications, where the results observed may have been previously misinterpreted or incomplete. However, there are some major concerns outlined below that should be addressed, which made the results and implications of their work more difficult to follow.

Major Comments:

The first concern is about Experiment 1 and 2. When the authors introduced the two different experiments, they state that Experiment 2 considered the confounding factors of Experiment 1. However, the differences between the two are not clearly described in methods, and the discussion should include a larger explanation of the specific confounding factors. Without this understanding and based on the "confounding factors" statement, it almost seems Experiment 1 should be part of Appendix, as there is a fair amount of results to track between the two experiments. Based on the two designs, I gathered that the second experiment aimed to allow for arm swing which could have possibly encouraged larger amounts of transverse pelvic rotation. Assuming that is true, discussion of the decoupled result of increased transverse pelvic rotation without a significant difference in arm swing should be provided. If this was not the motivation of Experiment 2, the authors should provide what confounding factors they hoped to account for and if they were successful in doing so. In addition, although both the experiments aimed to

provide additional degrees of freedom, they were not successful in allowing for the motion. Can the authors comment on the possible reasons that led to the motion still being constrained?

The second concern is about expanding on the implications of the results. In the discussion, the authors suggested that other studies must take into account the additional constraints elicited from the lateral stabilization. This is an interesting statement and is one of the main reasons for this study. However, the implications of the additional constraints are not fully addressed and need to be further explored. Specifically, how does the direct or indirect effects of lateral stabilization could lead to (or influence) the energetic results of the experiments in this study and previous studies?

As the aim of the device is to provide lateral stabilization, can the authors comments on whether or not they believe that the additional effects observed may be a result of the stabilization rather than the physical constraints of apparatus? For example, it has previously been thought that the reduction in step-width is a result of the lateral stabilization as the stepping strategy is supplemented. Could the same be true for the frontal pelvis rotation, supplementing the trunk strategy for maintaining stability? This may also provide an explanation as to why when the additional degree of freedom was included in Experiment 1, additional pelvis rotation was not observed.

The third concern is that, based on the discussion and the results, there is some lack of clarity on some of the results of the study. This is partially due to the lack of clarity between Experiment 1 and Experiment 2, but I believe further clarification is required to make sure the results are clearly communicated.

Page 14, Line 19-22: Please specify which experiment this result was obtained for. Additionally, an increase transverse and frontal plane pelvis rotation was reported, but according to Table 1 and 2, only the second experiment led to a significant increase in the transverse pelvic rotation of comparing the free and restricted conditions. There does not appear to be any significant difference in the frontal pelvis rotation between the restricted and free conditions. I am unsure if my understanding of results is incorrect or if this statement is incorrect.

Page 15, Line 33 to 38: Was there any statistical analysis between the two different experiments? It does appear that the amplitude is higher looking at Fig. 3 C and D, but it would be helpful to specify whether the statement is qualitative or quantitative.

Slight modifications to the figures and tables would greatly improve that readability of the paper.

Figure 2: The degrees of freedom of the two different apparatuses are a little unclear in the images. Adding arrows to show the motion of the degrees of freedom about the one or two axes of rotation would enhance the readability of the image.

Table 1 & 2: A caption for these two tables would make the manuscript easier to read so the reader can keep track of which table corresponds to which set of experiments.

Figure 3: The caption should specify that it is angular limb kinematics.

Figure 4: The caption should be more specific to the figure saying group average pelvis displacements.

Figures 3-7: Adding asterisks to show a significant difference between conditions of the same speed would make the results of the analysis much clearer.

There are some clarifications needed for the methods section as well.

Could the authors comment on what they hoped to accomplish by conducting the experiment at multiple speeds?

Page 10, Line 3: Were the displacements normalized at all to either the subject height or leg length?

I believe that certain subjects brought up in the introduction of the paper need to be revisited or further supported.

Page 5, Line 6-8: The definition of gait stability provided is not necessarily true as gait does not have one perfect measure (the manuscript's reference 2 explores this in depth); where many measures are not just about the base of support and centre of mass. It might be better to qualify the gait stability as achieved with interactions between the base of support and centre of mass rather than defining it strictly.

Minor Comment: Page 5, Line 27: Lateral does not need to be capitalized.

Reviewer: 2

Comments to the Author(s)

The objective of the current study was to determine whether previously employed methods to confer external lateral stabilization also reduce transverse plane rotation of the pelvis, which could confound the interpretation of previous studies (namely on medial-lateral COM displacement and metabolic cost).

It is important to note that the the cited methods of external lateral stabilization are primarily used to examine lateral balance control (i.e., medial-lateral COM displacement and foot placement control) and associated the associated metabolic cost, the indirect measures described in the current study. The premise is that by providing external lateral stabilization, study participants won't have to control medial-lateral step to step balance, allowing an assessment of the metabolic cost associated with medial-lateral step-to-step balance control.

Novel methods are introduced to both restrict and allow free transverse plane rotation of the pelvis. There are a number of interesting results, yet the main motivation for the study, limiting transverse plane pelvic rotation could explain the decrease in ML COM displacement, need for foot placement control, and the associated metabolic cost, are not addressed in the discussion or conclusion. A number of other concerns require attention as well.

1. My primary concern is that the main motivation for the study as presented in the introduction (i.e., traditional methods of external lateral stabilization may also limit transverse plane pelvic motion, which may in turn confound the interpretation of previous studies using these methods to investigate key aspects of medial-lateral balance control (COM displacement, foot placement control, and metabolic cost) is not addressed at all. In fact, the results suggest the opposite, that transverse plane pelvic rotation has little or no effect on these three metrics. Thus, it would appear that the existing interpretation of those studies stands. Despite being a major component to the motivation for the study, this point is not acknowledged nor discussed within the manuscript. Based on the introduction as written, this would seem to be the main result. As written, the last paragraph in the discussion ignores this key point, and goes on to suggest that in fact we should have concern when interpreting the results of those prior studies. The data presented do not seem to support those concerns and hence conclusions. Please revise.

2. It seems as though there is perhaps a missed opportunity to discuss what if any metabolic benefit exists to transverse pelvic rotation. This could be brought up in the context of the original determinants of gait (Kerrigan et al.)

a. Kerrigan DC, Riley PO, Lelas JL, Della Croce U. Quantification of pelvic rotation as a determinant of gait. *Arch Phys Med Rehabil* 2001;82:217-20.

b. Della Croce U, Riley PO, Lelas JL, Kerrigan DC. A refined view of the determinants of gait. *Gait and Posture* 2001;14(2):79-84.

3. Much of the discussion consists of rehashing the results. The discussion could be much more focused, thereby shortening the paper overall.

4. Please confirm the normality, or lack thereof, in the data (i.e., suitability of parametric tests). It is also odd to report medians but then use parametric tests. Please explain.

Minor

1. There are probably better references that could be used when citing ankle, foot placement, and hip balance control strategies.

2. Plots in Figure 3 could probably be increased in size to improve visibility.

Reviewer: 3

Comments to the Author(s)

The authors used a device, in two variants, to provide elastic support to participants during walking. Participants wore a waist belt attached to a frame that was connected to lateral supports by springs mounted on horizontal sliders. Two experiments tested two different frames that differed in allowing or constraining rotation in the frontal plane. Both frames allowed to either lock or release a constraint on rotation in the transverse plane. The main goal was to investigate the effects of lateral stabilization on mechanical and metabolic gait features, and whether the constraint in the transverse plane makes a difference. The results do show an effect of stabilization on some gait parameters, but not others that other studies found affected, most notably energy cost. Releasing the transverse constraint increased the rotation in that direction, but not up to the level during normal walking.

This study is one incremental step in a larger body of work investigating lateral mechanical stabilization during walking. One problem with lateral stabilization is to design a device that constrains some degrees of freedom in the desired fashion, while leaving others unconstrained. This study makes some interesting contributions to this field, but suffers from presenting these in a way that I found somewhat hard to follow, which might be a symptom of an underlying lack of clarity in the experimental design.

The main goal of this study is to compare the effect of lateral stabilization with and without a constraint on pelvis rotations in the transverse plane, implemented by locking or releasing lateral sliders that allow the connection point between the pelvis and the stabilization frame on either side to move in the anterior-posterior direction, labelled "free" and "restricted" walking. This is compared with a third condition of "normal" walking, without any stabilization. The authors performed two experiments, where Experiment 2 differed from Experiment 1 by (a) adding a two-minute familiarization period for each condition and (b) using a mechanically simpler stabilization device that removed the outer frame, thus making the device lighter, but also constraining pelvis rotation in the frontal plane. I don't really understand the rationale for two different experiments here. The authors state that "Experiment 2 supplemented experiment 1, as

it considered several potential confounding factors in the design and set-up of experiment 1” – but these confounding factors are not listed or discussed in detail. The main difference is the removal of the outer frame in the device, thus constraining pelvis rotations in the frontal plane. This seems to *add* a potential confounding factor, rather than removing it. I can see how the absence of a familiarization period in Experiment 1 is a potential confounding factor, but on its own, I wouldn't say that provides sufficient rationale for a whole new experiment. Could you please elaborate in what manner the addition of the frontal plane constraint removes a confounding factor, and what other confounding factors that Experiment 2 avoided?

Another factor is that in the device used here, the connection between the body-worn frame and the external anchoring frame was not fixed, but mounted on trolleys that were free to move in the anterior-posterior direction. This, the authors argue, removes a back door for using the lateral springs to generate forward-pulling forces that decrease the metabolic cost of walking. This is a very interesting point, because this would imply that the results from other experiments showing that lateral stabilization reduced metabolic cost might be spurious results of the forces in the anterior-posterior direction implicitly generated by the mechanism. This point is only made in the discussion, maybe because the authors also only realized this after seeing their results? If this was something the authors were aware of before conducting the experiment, I would suggest adding this to the list of research questions they attempted to answer. Otherwise, I suggest that you clarify that this result was surprising and the explanation was generated post-hoc. In the latter case, I would also recommend bringing this up a bit later in the Discussion section: currently it is the first phenomenon that is discussed in-depth, after the general first paragraph, suggesting that this is the main result, which is at odds with it not being mentioned as a research question earlier.

The authors made all data and analysis scripts available, which is great. Looking over what's in this cloud drive, however, I noticed some weird issues. One issue is that some of the variables have very large jumps, as seen in software/Plots, for example the Right Arm Swing in Subject 1, Trial 1 or Subject 14, Trial 5. This might be a problem with calculating angles from the rigid body orientation given by three markers on the cluster, since the jumps seem to be roughly around 90deg. Similar jumps occur in ML Pelvis Displacement, though, e.g. Subject 1, Trial 6; Subject 5, Trial 9; Subject 9, Trial 4. Another issue is gaps in the data, where some of the trajectories will just disappear for some of the gait cycle, e.g. in Right Arm Swing in Subject 10, Trial 9 or Transverse Pelvis Rotation Subject 1, Trial 4, where around 60% of the gait cycle *all* data is missing, similarly for Subject 6, Trial 5 around 0-20%. I did not go through the analysis code in detail, so it is possible that these are just intermediate results, before such issues have been weeded out by the authors, although the readme.docx seems to suggest that this is not the case. If this is the case and these artifacts are still part of the data as analyzed in the manuscript, then I suggest that the authors go back to the data processing stage and take a close and careful look at where they come from and how to avoid them. In some cases, removing a small number of problematic gait cycles might be sufficient, but in other cases, all data seems to be missing for part of the gait cycle, and I don't know of a good way to deal with this.

Minor issues and questions

p.7, l.32 “the pelvis was restricted from rotating in the transverse plane” – what exactly was the mechanism here to release or lock this constraint? Figure 2 seems to imply that some slider can be locked. Please add some detail here.

p.7, l.45 “normal walking” – did normal walking consist of walking while wearing the device but without lateral springs attached, or of walking entirely without the device? Please add this information to the text. If the latter, did you confirm that the normal arm swing was not impeded by the frame at all? The hands can reach quite far ahead of the body in normal arm swing, especially at fast walking speeds, and from Figures 1 and 2 it seems that arm swing might be impeded.

p.9, l.8 “Clusters of three infrared markers were attached to ... the left and right arms” – please specify where exactly on the arms the clusters were placed.

p.9, l.29/50 Please specify which convention you used for axes of rotation in calculating Euler angles.

p.9, l.46 – how did you identify the heel strike events?

p.10, l.10f: anterior-posterior pelvis displacement – at this point it seems odd that the ap-displacement is defined differently than the other displacements/angles. Is this because the ap-displacement is relevant in terms of the implicit forces in this direction that were *not* applied by your device due to the trolley connection? If so, it would help to explain that at this point, and maybe give this variable a different name to more clearly differentiate it from the other means.

p.10, l.35 “distances between ... foot placements” – what exactly was the location of the foot placement here?

p.10, l.37 “for the step length, we calculated the average over legs, since nonsignificant differences were found between left and right step lengths” – what about step width? If it was not significant, did you also pool data? If it was significant, did you use it as a factor in the statistical analysis?

Most figures: The legend indicates that the whiskers of the box-and-whisker plots cover the whole range of the data, from “Min” up to “Max”. However, many of the individual data points are outside of this range, so that cannot be correct. Please clarify. Also, it would help to add some horizontal jitter to the individual data points, so they are still distinguishable when multiple data points are close together.

Language

p.6, l.1 constrains → constraints

p.9, l.55 matrixes → matrices

multiple locations: capitalize “Experiment 1”

– Hendrik Reimann

Author's Response to Decision Letter for (RSOS-200647.R0)

See Appendix A.

RSOS-202088.R0

Review form: Reviewer 1

Is the manuscript scientifically sound in its present form?

Yes

Are the interpretations and conclusions justified by the results?

Yes

Is the language acceptable?

Yes

Do you have any ethical concerns with this paper?

No

Have you any concerns about statistical analyses in this paper?

No

Recommendation?

Accept with minor revision (please list in comments)

Comments to the Author(s)

This manuscript demonstrated that the typical methods used to provide lateral stabilization during gait provides additional effects aside from improving mediolateral stability. To remedy this issue, the authors presented two novel designs that aim to allow for frontal and transverse pelvis rotation that were restricted in previous studies. The first design had two additional degrees of freedom to allow for frontal and transverse pelvis rotation, while the second removed the frame that allowed for frontal pelvis rotation to reduce the weight of the device. The first experiment showed that no change in frontal pelvis rotation but also no change between free and restricted conditions for transverse pelvis rotation. The second experiment found reduced frontal pelvis rotation and significantly greater transverse pelvis rotation for free over the restricted condition, although it was still lower than normal walking.

Overall, this paper does an excellent job of describing the effect of mediolateral stabilization and the two novel designs presented offered intuitive solutions for observed issues. Based on the provided author responses to reviewers, the authors have thoroughly incorporated feedback from the previous review, and the manuscript has greatly improved. In my opinion, I believe that this manuscript should be accepted, but I do have a few comments that I believe would enhance the paper.

Major Comments:

Page 14 Line 15: The "normal" frontal pelvis rotations refers to no significant difference among Normal, Free, and Restricted for Experiment 1, but the "more normal" transverse pelvis rotation for Experiment 2 is because Free is significantly greater than Restricted. One might misinterpret the latter as no significant difference among the three conditions, although the wording "more normal" helps. In addition, it is ambiguous whether "the provided rotational degrees of freedom" refers to Free or Restricted or both. I suggest an amendment to help clarify the sentence: "... allowing free transverse pelvis rotation in our new set-ups resulted in normal frontal pelvis rotation (Experiment 1), or more normal (i.e. greater) transverse plane pelvis rotation (Experiment 2)..."

Page 14, Line 18: I think it would be helpful to be more specific which experiment is being referred to in this line as here was no observed reduction in the frontal pelvis rotation in Experiment 1 (per figure 3). Extending from this point, I would encourage the authors to comment on why there was no significant difference in the frontal pelvis rotation for Experiment 1 in the restricted condition. Based on the information provided in the introduction, I would have expected there to be a reduction, especially in the restricted condition.

Page 14, Line 23: In this section, the authors suggest that the reduction in the vertical center of mass displacement could be responsible for the reduction in metabolic cost. Although valid references are given to support this statement, the effect of vertical centre of mass displacement still quite disputed throughout the literature (see references below). To be consistent with past literature, it would be helpful include that reduced vertical displacement could be one cause of reduced metabolic cost, but since there is no consensus on the effect of the reduced centre of mass displacement, there may be additional causes that reduce metabolic cost.

References:

Keith E. Gordon, Daniel P. Ferris, Arthur D. Kuo; Metabolic and Mechanical Energy Costs of Reducing Vertical Center of Mass Movement During Gait; Archives of Physical Medicine and Rehabilitation; Volume 90, Issue 1; 2009; Pages 136-144,

Justus D. Ortega and Claire T. Farley; Minimizing center of mass vertical movement increases metabolic cost in walking; Journal of Applied Physiology 2005 99:6, 2099-2107

Minor Comments:

Page 4 Line 19: Please include if transverse pelvis rotation increases or decreases step length.

Review form: Reviewer 2

Is the manuscript scientifically sound in its present form?

Yes

Are the interpretations and conclusions justified by the results?

Yes

Is the language acceptable?

Yes

Do you have any ethical concerns with this paper?

No

Have you any concerns about statistical analyses in this paper?

No

Recommendation?

Accept as is

Comments to the Author(s)

The authors have addressed my primary concerns.

Decision letter (RSOS-202088.R0)

Dear Dr Mahaki

On behalf of the Editors, we are pleased to inform you that your Manuscript RSOS-202088 "How does external lateral stabilization constrain normal gait, apart from improving medio-lateral gait stability?" has been accepted for publication in Royal Society Open Science subject to minor

revision in accordance with the referees' reports. Please find the referees' comments along with any feedback from the Editors below my signature.

Please submit your revised manuscript and required files (see below) no later than 7 days from today's (ie 17-Feb-2021) date. Note: the ScholarOne system will 'lock' if submission of the revision is attempted 7 or more days after the deadline. If you do not think you will be able to meet this deadline please contact the editorial office immediately.

on behalf of Dr Manoj Srinivasan (Associate Editor) and Pietro Cicuta (Subject Editor)
openscience@royalsociety.org

Associate Editor Comments to Author (Dr Manoj Srinivasan):

The reviewers were satisfied with your revisions in response to their comments. One of the reviewers has provided some additional comments, all of which can be addressed with small edits (as suggested by the reviewer). We look forward to a revised manuscript.

Reviewer comments to Author:

Reviewer: 1
Comments to the Author(s)

This manuscript demonstrated that the typical methods used to provide lateral stabilization during gait provides additional effects aside from improving mediolateral stability. To remedy this issue, the authors presented two novel designs that aim to allow for frontal and transverse pelvis rotation that were restricted in previous studies. The first design had two additional degrees of freedom to allow for frontal and transverse pelvis rotation, while the second removed the frame that allowed for frontal pelvis rotation to reduce the weight of the device. The first experiment showed that no change in frontal pelvis rotation but also no change between free and restricted conditions for transverse pelvis rotation. The second experiment found reduced frontal pelvis rotation and significantly greater transverse pelvis rotation for free over the restricted condition, although it was still lower than normal walking.

Overall, this paper does an excellent job of describing the effect of mediolateral stabilization and the two novel designs presented offered intuitive solutions for observed issues. Based on the provided author responses to reviewers, the authors have thoroughly incorporated feedback from the previous review, and the manuscript has greatly improved. In my opinion, I believe that this manuscript should be accepted, but I do have a few comments that I believe would enhance the paper.

Major Comments:

Page 14 Line 15: The "normal" frontal pelvis rotations refers to no significant difference among Normal, Free, and Restricted for Experiment 1, but the "more normal" transverse pelvis rotation for Experiment 2 is because Free is significantly greater than Restricted. One might misinterpret the latter as no significant difference among the three conditions, although the wording "more normal" helps. In addition, it is ambiguous whether "the provided rotational degrees of freedom" refers to Free or Restricted or both. I suggest an amendment to help clarify the sentence: "... allowing free transverse pelvis rotation in our new set-ups resulted in normal frontal pelvis rotation (Experiment 1), or more normal (i.e. greater) transverse plane pelvis rotation (Experiment 2)..."

Page 14, Line 18: I think it would be helpful to be more specific which experiment is being referred to in this line as here was no observed reduction in the frontal pelvis rotation in Experiment 1 (per figure 3). Extending from this point, I would encourage the authors to comment on why there was no significant difference in the frontal pelvis rotation for Experiment 1 in the restricted condition. Based on the information provided in the introduction, I would have expected there to be a reduction, especially in the restricted condition.

Page 14, Line 23: In this section, the authors suggest that the reduction in the vertical center of mass displacement could be responsible for the reduction in metabolic cost. Although valid references are given to support this statement, the effect of vertical centre of mass displacement still quite disputed throughout the literature (see references below). To be consistent with past literature, it would be helpful include that reduced vertical displacement could be one cause of reduced metabolic cost, but since there is no consensus on the effect of the reduced centre of mass displacement, there may be additional causes that reduce metabolic cost.

References:

Keith E. Gordon, Daniel P. Ferris, Arthur D. Kuo; Metabolic and Mechanical Energy Costs of Reducing Vertical Center of Mass Movement During Gait; Archives of Physical Medicine and Rehabilitation; Volume 90, Issue 1; 2009; Pages 136-144,

Justus D. Ortega and Claire T. Farley; Minimizing center of mass vertical movement increases metabolic cost in walking; Journal of Applied Physiology 2005 99:6, 2099-2107

Minor Comments:

Page 4 Line 19: Please include if transverse pelvis rotation increases or decreases step length.

Reviewer: 2

Comments to the Author(s)

The authors have addressed my primary concerns.

===PREPARING YOUR MANUSCRIPT===

===PREPARING YOUR REVISION IN SCHOLARONE===

<https://royalsociety.org/journals/authors/author-guidelines/#supplementary-material> to include a suitable title and informative caption. An example of appropriate titling and captioning may be found at https://figshare.com/articles/Table_S2_from_Is_there_a_trade-off_between_peak_performance_and_performance_breadth_across_temperatures_for_aerobic_scops_in_teleost_fishes_/3843624.

Author's Response to Decision Letter for (RSOS-202088.R0)

See Appendix B.

Decision letter (RSOS-202088.R1)

Dear Dr Mahaki,

It is a pleasure to accept your manuscript entitled "How does external lateral stabilization constrain normal gait, apart from improving medio-lateral gait stability?" in its current form for

publication in Royal Society Open Science. The comments of the reviewer(s) who reviewed your manuscript are included at the foot of this letter.

on behalf of Dr Manoj Srinivasan (Associate Editor) and Pietro Cicuta (Subject Editor)
openscience@royalsociety.org

Appendix A

Reviewers' Comments to Author:

Reviewer 1

Comments to the Author(s)

This paper demonstrates that external lateral stabilization constrains normal gait, providing effects aside from improving mediolateral gait stability. Furthermore, this paper proposes and tests two novel designs for a new apparatus for mediolateral stabilization to negate the unwanted effects seen in previous experiments. Specifically, previous mediolateral stabilization apparatuses directly led to a reduction in transverse and frontal pelvis rotations and vertical pelvis displacement. The first proposed design had additional degrees of freedom to allow for frontal and transverse pelvis rotation, whereas the second design only had one additional degrees of freedom to allow for transverse pelvis rotation. However, the smaller frame of the second design was intended to unconstrain arm swing. The two designs were tested separately with two different subject groups in three conditions (normal, free, and restricted) at three different speeds. The first design did not result in any significant differences in frontal and transverse rotation of the pelvis between the free and restricted conditions. However, both conditions significantly reduced pelvis rotation and mediolateral displacements compared to the normal condition. Indirectly, the mediolateral stabilization led to reduced transverse thorax rotation, arm swing, and step width. The results of the second design were similar, however, the free condition resulted in a more transverse pelvis rotation than the restricted condition but not as much as seen in the normal condition. The paper concluded that more elaborate set-ups are required to provide mediolateral stability without constraining other aspects of gait.

This paper thoroughly described and analyzed the changes that were elicited by the mediolateral stabilization. The two designs proposed in this study were novel yet very intuitive that would appear to provide a possible solution to the issues with transverse and frontal pelvis rotation. Additionally, this paper takes the opportunity to explore the results of past publications, where the results observed may have been previously misinterpreted or incomplete. However, there are some major concerns outlined below that should be addressed, which made the results and implications of their work more difficult to follow.

We wish to thank the reviewer for their comments, which we feel really helped improve the manuscript. Below, we provide a point-by-point reply, and any changes made in the manuscript have been marked. We would like to mention explicitly here that based on a comment of Reviewer 3, we have reanalyzed our data (it contained some irregularities; the advantage of sharing data and code was that this could be picked up at this point by the reviewer, which prevented us from publishing wrong results), which has led to slightly different results than the previous version of the manuscript.

Major Comments:

1. The first concern is about Experiment 1 and 2. When the authors introduced the two different experiments, they state that Experiment 2 considered the confounding factors of Experiment 1. However, the differences between the two are not clearly described in methods, and the discussion should include a larger explanation of the specific confounding factors. Without this understanding and based on the "confounding factors" statement, it almost seems Experiment 1 should be part of Appendix, as there is a fair amount of results to track between the two experiments. Based on the two designs, I gathered that the second experiment aimed to allow for arm swing which could have possibly encouraged larger amounts of transverse pelvic rotation. Assuming that is true, discussion of the decoupled result of increased transverse pelvic rotation without a significant difference in arm swing should be provided. If this was not the motivation of Experiment 2, the authors should provide what confounding factors they hoped to account for and if they were successful in doing so. In addition, although both the experiments aimed to provide additional degrees of freedom, they were not successful in allowing for the motion. Can the authors comment on the possible reasons that led to the motion still being constrained?

We have tried to more clearly describe the differences between the two experiments and to explain about the motivation of Experiment 2. Specifically, we have provided following sentences in the Methods and Discussion:

Methods, page 5, lines 15-22 and page 6, line 1:

“Experiment 1 was performed to test the effect of external lateral stabilization with and without constrained transverse pelvic rotation on mechanical and metabolic gait features (Figure 1. & Figure 2. A). However, the results of Experiment 1 failed to reach the significant reduction of energy cost in the stabilized condition which was reported by some previous studies [1-3]. The potential effects of the frame weight on energy cost was considered as a potential confounding factor of Experiment 1. Additionally, the lack of habituation time to allow the participants for full familiarization with the set-up was considered as another reason for our inability to reduce energy cost in stabilized condition. Having the same aim and taking these potential confounding factors (weight of frame and habituation time) into account, we performed Experiment 2 to supplement Experiment 1.”

Methods, page 7, lines 20-21:

“The frame in Experiment 2 had no outer frame (Figure 2. B), and thus had a reduced weight (weight =1.5 kg) compared to the frame used in Experiment 1.”

Methods, page 8, lines 4-5:

“The protocol for Experiment 2 was equal to Experiment 1, except that participants were familiarized with walking in each condition for about 2 minutes.”

In the Experiment 2, our results again failed to reach a significant reduction of energy cost in stabilized conditions. We have explained the potential reasons of this as follows:

Discussion, page 17, lines 10-21:

“Thirdly, providing more than 30 minutes habituation time, Ortega et al. [3] reported a significant 3-4% reduction of energy cost in stabilized condition. However, we did not provide any habituation time in the Experiment 1 which was considered as a potential confounding factor. Therefore, we performed the Experiment 2 to allow the participants to become more familiarize with walking in stabilized conditions. Compared to Ortega et al. [3], a shorter habituation time (i.e. between 3-10 minutes) was used in our Experiment 2 and in the previous studies [1, 4] with the reported nonsignificant effect of lateral stabilization on energy cost. Therefore, a too short habituation time to allow the participants for full familiarization might be responsible for our inability to reduce energy cost in stabilized conditions. Lastly, although the potential effects of the frame weight on energy cost were minimized by removing outer frame in the Experiment 2, the frames used by IJmker et al. [5, 4] and our two experiments may have added additional energy cost and subsequently these factors might offset some of the reduced energy cost induced by external lateral stabilization.”

2. The second concern is about expanding on the implications of the results. In the discussion, the authors suggested that other studies must take into account the additional constraints elicited from the lateral stabilization. This is an interesting statement and is one of the main reasons for this study. However, the implications of the additional constraints are not fully addressed and need to be further explored. Specifically, how does the direct or indirect effects of lateral stabilization could lead to (or influence) the energetic results of the experiments in this study and previous studies?

Here, we must admit that for most of these (in)direct effects, it's not immediately clear how they would influence energetic cost, but it is clear that they could. For instance, horizontal pelvis rotation, as one of the gait determinants [6], reduces vertical pelvis displacement and subsequently reduces energy cost [7]. Therefore, reduction of horizontal pelvis rotation might offset these mechanical and metabolic benefits. Moreover, the reduction of vertical pelvis displacement might be due to downward/upward forces induced by lateral stabilization. We have tried to more explicitly state how we think (in)direct effects of lateral stabilization influence energetic cost, both in the introduction and discussion as follows:

Introduction, page 5, lines 1-2:

“For example, the reduced energy cost in stabilized walking could be due to the aforementioned gait pattern constraints, rather than a reduced need to control medio-lateral gait stability [1-3].”

Discussion, page 14, lines 18-23 and page 15, lines 1-4:

“The reduced frontal and transverse pelvic rotations might confound the interpretation of previously reported results. For instance, our new set-ups resulted in normal frontal and more transverse pelvis rotations. Therefore, the reduced frontal and transverse pelvis rotations need to be considered as the results of the physical constraints of apparatus and cannot be attributed to a strategy to control gait stability. Moreover, it has been reported that transverse pelvis rotation, as one of the gait determinants [6], has some mechanical and metabolic benefits as it reduces vertical center of mass displacement [8, 6] and contributes to step length [8]. Therefore, restricted transverse pelvis rotation induced by lateral stabilization might offset these benefits. In line with this, the restricted step length in our Experiment 1 could be due to the restricted transverse pelvis rotation and unaffected step length in our Experiment 2 could be considered as the mechanical benefits of increased transverse pelvis rotation.”

Discussion, page 15, lines 5-12:

“The reduced vertical pelvis displacement could be due to downward/upward forces induced by lateral stabilization which might also confound the interpretation of reported results. If bilateral springs provide downward forces, subjects might need some extra energy cost to compensate them, which consequently might offset the energy cost savings of external lateral stabilization. Conversely, the potential upward forces induced by springs might decrease the body weight which could increase energy cost savings of lateral stabilization. To avoid downward/upward forces, one possibility is to use longer rope to attach the springs to the frame. An alternative possibility may be to attach the springs to rails that can also move in the vertical direction, although this may also induce unwanted oscillations.”

Discussion, page..., lines...:

Page 17, lines 4-6:

“Firstly, springs that are fixed in anterior-posterior direction as used by previous studies [2, 3] might provide unwanted forces and assistance in the anterior-posterior direction and might have increased some of the energy cost savings of external lateral stabilization.”

3. As the aim of the device is to provide lateral stabilization, can the authors comments on whether or not they believe that the additional effects observed may be a result of the stabilization rather than the physical constraints of apparatus? For example, it has previously been thought that the reduction in step-width is a result of the lateral stabilization as the stepping

strategy is supplemented. Could the same be true for the frontal pelvis rotation, supplementing the trunk strategy for maintaining stability? This may also provide an explanation as to why when the additional degree of freedom was included in Experiment 1, additional pelvis rotation was not observed.

Based on the first comment of Reviewer 3, we have reanalyzed our data and we ended up with a slightly different result for the frontal pelvis rotation in Experiment 1. Our new results showed that frontal pelvis rotation was not restricted by the frame used in Experiment 1, however it was restricted by the frame used in Experiment 2 in which we removed the free degree of motion in frontal plane between pelvis and frame. Thus, it can be concluded that the reduced frontal and horizontal pelvis rotations are because of physical constrain of the frame and they cannot be considered a results of stabilization. This issue has been addressed here:

Page 14, lines 18-22:

“The reduced frontal and transverse pelvic rotations might confound the interpretation of previously reported results. For instance, the provided rotational degrees of freedom in our new set-ups resulted in normal frontal and more transverse pelvis rotations. Therefore, the reduced frontal and transverse pelvis rotations induced by lateral stabilization need to be considered as the results of the physical constraints of apparatus and cannot be attributed to a strategy to control gait stability.”

4. The third concern is that, based on the discussion and the results, there is some lack of clarity on some of the results of the study. This is partially due to the lack of clarity between Experiment 1 and Experiment 2, but I believe further clarification is required to make sure the results are clearly communicated.

We hope that our answer in response to your first question has resolved this issue.

5. Page 14, Line 19-22: Please specify which experiment this result was obtained for. Additionally, an increase transverse and frontal plane pelvis rotation was reported, but according to Table 1 and 2, only the second experiment led to a significant increase in the transverse pelvic rotation of comparing the free and restricted conditions. There does not appear to be any significant difference in the frontal pelvis rotation between the restricted and free conditions. I am unsure if my understanding of results is incorrect or if this statement is incorrect.

We agree with the reviewer that the results may have been unclear here, and have changed that text to be more explicit about which experiment we are talking:

Page 14, lines 3-5:

“The removal of pelvic rotation constraints in our experimental set-up led to increased frontal pelvis rotation in Experiment 1 and also increased transverse pelvis rotation in Experiment 2.”

6. Page 15, Line 33 to 38: Was there any statistical analysis between the two different experiments? It does appear that the amplitude is higher looking at Fig. 3 C and D, but it would be helpful to specify whether the statement is qualitative or quantitative.

There was no statistical comparison between the experiments, because participants, study set-ups and protocols were different between two experiments. To make this more clear, we now explicitly state this in the statistics section.

Page 10, lines 18:

“We did not perform statistical comparisons between Experiments 1 and 2 since the participants, study set-ups and protocols were different between two experiments.”

We have revised these lines as follows:

Page 14, lines 9-17:

“Our results also showed that frontal plane pelvis rotation was not reduced by lateral stabilization with the frame used in Experiment 1. The frame used in Experiment 1 included an inner and an outer frames, which were attached to each other and provided a free rotational degree of motion in the frontal plane between pelvis and frame, however the frame used in Experiment 2 did not have an outer frame and it was more similar to the frame used by previous studies [5, 4]. Therefore, in contrast to the set-ups used by previous studies in which frontal and transverse pelvis rotations were restricted, the provided rotational degrees of freedom in our new set-ups resulted in normal frontal pelvis rotation (Experiment 1), or more normal transverse plane pelvis rotation (Experiment 2), but not both at the same time.”

7. Slight modifications to the figures and tables would greatly improve that readability of the paper.

Figure 2: The degrees of freedom of the two different apparatuses are a little unclear in the images. Adding arrows to show the motion of the degrees of freedom about the one or two axes of rotation would enhance the readability of the image.

We have improved the figure resolution. We have added arrows to indicate the degrees of freedom together with related axes in this figure, the new figure appears below:

Fig 2. Schematic representation of the lateral stabilization frames used in Experiments 1 (A) and 2 (B). (1) waist belt (we attached kinematic markers here); (2) inner frame which moves inside the outer frame to allow frontal pelvic rotation in Experiment 1; (3) slider between waist belt and inner frame to allow transverse pelvic rotation; (4) Two screws resisting the sliders to work (5) outer frame allowing normal arm swing (6) the joint in which inner and outer frame were attached to each other and provided a free rotational degree of motion in the frontal plane. Arrows show the degrees of freedom around the related axes.

8. Table 1 & 2: A caption for these two tables would make the manuscript easier to read so the reader can keep track of which table corresponds to which set of experiments.

We have added following captions for Table 1. & 2:

Table 1. The direct and indirect effects of external lateral stabilization on gait features in **Experiment 1.**

Table 2. The direct and indirect effects of external lateral stabilization on gait features in **Experiment 2.**

9. Figure 3: The caption should specify that it is angular limb kinematics.

We have revised this caption as follows:

*“Figure 3. Group average transverse pelvis rotation (A & B) and frontal pelvis rotation (C & D) as well as transverse thorax rotation (E & F) (median \pm 25th percentile) at each walking speed in the Normal (black), stabilized conditions without transverse pelvis rotation restriction (blue) and with transverse pelvis rotation restriction (red) in Experiments 1 and 2. * denote significant differences between conditions (based on the results of Bonferroni correction of Condition effect). Individual data are also plotted as dots.”*

10. Figure 4: The caption should be more specific to the figure saying group average pelvis displacements.

We have revised this caption as follows:

*“Figure 4. Group average pelvis displacements (median \pm 25th percentile) at each walking speed in the Normal (black), stabilized conditions without transverse pelvis rotation restriction (blue) and with transverse pelvis rotation restriction (red) for medio-lateral (A & B), anterior-posterior (C & D), and vertical pelvis displacements (E & F) in experiments 1 and 2. * denote the significant difference between conditions (based on the results of Bonferroni correction of Condition effect). Individual data are also plotted as dots.”*

11. Figures 3-7: Adding asterisks to show a significant difference between conditions of the same speed would make the results of the analysis much clearer.

We have added * to show the significant difference between conditions based on the results of Bonferroni correction of Condition effect.

12. There are some clarifications needed for the methods section as well. Could the authors comment on what they hoped to accomplish by conducting the experiment at multiple speeds?

Generating the different levels of pelvis rotation, providing a greater generalizability, and increasing the statistical power were the reasons to conduct our experiments at multiple speeds. We have now made this more explicit in the manuscript:

Page 7, lines 7-9:

“To generate the different levels of transverse pelvis rotation, all conditions were executed at 3 speeds (0.83, 1.25, and 1.66 m/s). Measuring at multiple speeds also has the added advantage of greater generalizability and increased statistical power.”

13. Page 10, Line 3: Were the displacements normalized at all to either the subject height or leg length?

Since the study design was within-subject in which subjects were compared to themselves, the displacements were not normalized to the subject height and leg length.

14. I believe that certain subjects brought up in the introduction of the paper need to be revisited or further supported. Page 5, Line 6-8: The definition of gait stability provided is not necessarily true as gait does not have one perfect measure (the manuscript’s reference 2 explores this in depth); where many measures are not just about the base of support and center of mass. It might be better to qualify the gait stability as achieved with interactions between the base of support and center of mass rather than defining it strictly.

The reviewer is of course correct here. For instance, Lyapunov exponent as a measure of gait stability is independent of base of support. Thus, we have revised the definition of gait stability as suggested by the reviewer:

Page 4, lines 2-3:

“Gait stability is achieved by interactions between the base of support and body center of mass in the face of perturbations [9, 10].”

Minor Comment:

1. Page 5, Line 27: Lateral does not need to be capitalized.

We have removed the capitalization.

Reviewer 2

Comments to the Author(s)

The objective of the current study was to determine whether previously employed methods to confer external lateral stabilization also reduce transverse plane rotation of the pelvis, which could confound the interpretation of previous studies (namely on medial-lateral COM displacement and metabolic cost).

It is important to note that the cited methods of external lateral stabilization are primarily used to examine lateral balance control (i.e., medial-lateral COM displacement and foot placement control) and associated the associated metabolic cost, the indirect measures described in the current study. The premise is that by providing external lateral stabilization, study participants won't have to control medial-lateral step to step balance, allowing an assessment of the metabolic cost associated with medial-lateral step-to-step balance control.

Novel methods are introduced to both restrict and allow free transverse plane rotation of the pelvis. There are a number of interesting results, yet the main motivation for the study, limiting transverse plane pelvic rotation could explain the decrease in ML COM displacement, need for foot placement control, and the associated metabolic cost, are not addressed in the discussion or conclusion. A number of other concerns require attention as well.

We wish to thank the reviewer for their comments, which we feel really helped improve the manuscript. Below, we provide a point-by-point reply, and any changes made in the manuscript have been marked. We would like to mention explicitly here that based on a comment of Reviewer 3, we have reanalyzed our data (it contained some irregularities; the advantage of sharing data and code was that this could be picked up at this point by the reviewer, which prevented us from publishing wrong results), which has led to slightly different results than the previous version of the manuscript.

Major Comments:

1. My primary concern is that the main motivation for the study as presented in the introduction (i.e., traditional methods of external lateral stabilization may also limit transverse plane pelvic motion, which may in turn confound the interpretation of previous studies using these methods to investigate key aspects of medial-lateral balance control (COM displacement, foot placement control, and metabolic cost) is not addressed at all. In fact, the results suggest the opposite, that transverse plane pelvic rotation has little or no effect on these three metrics. Thus, it would appear that the existing interpretation of those studies stands. Despite being a major component to the motivation for the study, this point is not acknowledged nor discussed within the manuscript. Based on the introduction as written, this would seem to be the main result. As written, the last paragraph in the discussion ignores this key point, and goes on to suggest that in fact we should have concern when interpreting the results of those prior studies. The data presented do not seem to support those concerns and hence conclusions. Please revise.

On the one hand, the reviewer is correct that our results do not show that restriction of pelvis motion during stabilized gait leads to changes in mediolateral COM displacement, and foot placement as compared to a condition in which the pelvis can rotate freely. Based on our results, one could even state that free or restricted pelvis rotation during stabilized walking has no effect on energetic cost (and thus, no effects on the energetic costs associated with stabilizing gait). For the latter of course, we should take into account that we failed to find the reported decrease in stabilized walking for all conditions, which somewhat limits the statements that we can make here. We agree that these ideas were not expressed clearly enough in our discussion. We have rewritten this part of the discussion to more correctly state our ideas.

2. It seems as though there is perhaps a missed opportunity to discuss what if any metabolic benefit exists to transverse pelvic rotation. This could be brought up in the context of the original determinants of gait (Kerrigan et al.)

a. Kerrigan DC, Riley PO, Lelas JL, Della Croce U. Quantification of pelvic rotation as a determinant of gait. Arch Phys Med Rehabil 2001;82:217-20.

b. Della Croce U, Riley PO, Lelas JL, Kerrigan DC. A refined view of the determinants of gait. Gait and Posture 2001;14(2):79-84.

We have added following sentence to discuss about the potential mechanical and metabolic benefits of transverse pelvis rotation:

Page 14, lines 22-23 and page 15, lines 1-4:

“Moreover, it has been reported that transverse pelvis rotation, as one of the gait determinants, has some mechanical and metabolic benefits as it reduces vertical center of

mass displacement [8, 6] and contributes to step length [11]. Therefore, restricted transverse pelvis rotation induced by lateral stabilization might offset these benefits. In line with this, the restricted step length in our Experiment 1 could be due to the restricted transverse pelvis rotation and unaffected step length in our Experiment 2 can be considered as the mechanical benefit of increased transverse pelvis rotation.”

3. Much of the discussion consists of rehashing the results. The discussion could be much more focused, thereby shortening the paper overall.

We have reworked the discussion to be less rephrasing of the results.

4. Please confirm the normality, or lack therefore, in the data (i.e., suitability of parametric tests). It is also odd to report medians but then use parametric tests. Please explain.

Using Shapiro-Wilk tests, we have checked the normality of the data per trial (Tables S1. and S2.).

Table S1. The p-values of Shapiro-Wilk tests in **Experiment 1**.

Variables	Normal			Free			Restricted		
	0.83 (m/s)	1.25 (m/s)	1.66 (m/s)	0.83 (m/s)	1.25 (m/s)	1.66 (m/s)	0.83 (m/s)	1.25 (m/s)	1.66 (m/s)
Transverse pelvis rotation	0.34	0.78	0.81	0.69	0.77	0.66	0.04	0.01	0.05
Frontal pelvis rotation	0.99	0.18	0.06	0.97	0.60	0.43	0.16	0.17	0.50
Medio-lateral pelvis displacement	0.44	0.26	0.53	0.36	0.21	0.99	0.35	0.26	0.24
Anterior-posterior pelvis displacement	0.45	0.24	0.01	0.13	0.33	0.003	0.90	0.04	0.02
Vertical pelvis displacement	0.80	0.69	0.28	0.14	0.59	0.98	0.24	0.54	0.85
Anterior-posterior drift over trial	0.46	0.18	0.38	0.32	0.07	0.59	0.97	0.82	0.79
Transverse thorax rotation	0.30	0.39	0.50	0.22	0.88	0.02	0.59	0.69	0.08
Arm swing	0.76	0.09	0.50	0.63	0.63	0.14	0.23	0.44	0.52
Step length	0.62	0.45	0.46	0.05	0.27	0.59	0.86	0.40	0.68
Step width	0.18	0.22	0.55	0.01	0.71	0.44	0.61	0.73	0.29
Energy cost	0.58	0.25	0.93	0.42	0.74	0.96	0.07	0.95	0.69

The results showed that most of the p-values of Shapiro-Wilk tests were greater than 0.05, confirming the normal distribution of our data and suitability of parametric tests.

Table S2. The p-values of Shapiro-Wilk tests in **Experiment 2**.

Variables	Normal			Free			Restricted		
	0.83 (m/s) 2	1.25 (m/s)	1.66 (m/s)	0.83 (m/s) 4	1.25 (m/s)	1.66 (m/s)	0.83 (m/s)	1.25 (m/s) 3	1.66 (m/s)
Transverse pelvis rotation	0.00	0.29	0.11	0.00	0.10	0.78	0.05	0.01	0.14
Frontal pelvis rotation	0.87	0.28	0.39	0.37	0.11	0.00	0.16	0.00	0.01
Medio-lateral pelvis displacement	0.99	0.16	0.77	0.01	0.47	0.16	0.11	0.79	0.51
Anterior-posterior pelvis displacement	0.08	0.27	0.52	0.61	0.63	0.14	0.93	0.40	0.03
Vertical pelvis displacement	0.76	0.04	0.28	0.68	0.54	0.85	0.95	0.33	0.68
Anterior-posterior drift over trial	0.62	0.27	0.65	0.80	0.58	0.78	0.43	0.54	0.81

Transverse thorax rotation	0.03	0.05	0.04	0.57	0.74	0.99	0.70	0.75	0.96
Arm swing	0.02	0.39	0.84	0.04	0.11	0.26	0.22	0.19	0.26
Step length	0.55	0.79	0.38	0.65	0.96	0.46	0.27	0.71	0.33
Step width	0.85	0.09	0.07	0.07	0.00 1	0.45	< 0.00 1	0.00 1	0.03
Energy cost	0.12	0.46	0.31	0.79	0.45	0.86	0.89	0.45	0.16

In our Experiment 2, our results showed that most of the p-values of the Shapiro-Wilk tests were greater than 0.05, indicating the normal distribution of the data.

As mentioned above, most of the p-values were greater than 0.05 for both experiments, indicating that all of the gait measures were approximately normally distributed. Moreover, repeated measures ANOVA is robust to violations of normality. Therefore, we are confident in our results based on the parametric tests (i.e. repeated measures ANOVA). Given that the not normal distribution of the data is the case, we have ended up with approximately the same results based on non-parametric tests. To clarify this issue, we have added following sentence in the statistical section:

Page 10, line 18:

“The Shapiro-Wilk tests confirmed the normal distribution of data for most of the trials ($p > 0.05$). Thus,....”

Minor comments

1. There are probably better references that could be used when citing ankle, foot placement, and hip balance control strategies.

We have added better references for citing aforementioned strategies. Specifically, we now cited:

Hof, A. L., van Bockel, R. M., Schoppen, T., Postema, K. 2007 Control of lateral balance in walking: experimental findings in normal subjects and above-knee amputees. *Gait Posture*. 25, 250-258.

Fettrow, T., Reimann, H., Grenet, D., Thompson, E., Crenshaw, J., Higginson, J., Jeka, J. 2019 Interdependence of balance mechanisms during bipedal locomotion. *Plos one*. 14, e0225902.

Fettrow, T., Reimann, H., Grenet, D., Thompson, E., Crenshaw, J., Higginson, J., Jeka, J. 2019 Interdependence of balance mechanisms during bipedal locomotion. *Plos one*. 14, e0225902.

Reimann, H., Fettrow, T., Thompson, E. D., Jeka, J. J. 2018 Neural control of balance during walking. *Frontiers in physiology*. 9,

Stimpson, K. H., Heitkamp, L. N., Horne, J. S., Dean, J. C. 2018 Effects of walking speed on the step-by-step control of step width. *J Biomech*. 68, 78-83.

Rankin, B. L., Buffo, S. K., Dean, J. C. 2014 A neuromechanical strategy for mediolateral foot placement in walking humans. *Journal of neurophysiology*. 112, 374-383.

Wang, Y., Srinivasan, M. 2014 Stepping in the direction of the fall: the next foot placement can be predicted from current upper body state in steady-state walking. *Biology letters*. 10, 20140405.

Arvin, M., Hoozemans, M., Pijnappels, M., Duysens, J., Verschueren, S. M. P., Van Dieen, J. 2018 Where to step? Contributions of stance leg muscle spindle afference to planning of mediolateral foot placement for balance control in young and older adults. *Frontiers in Physiology*. 9, 1134.

Hurt, C. P., Rosenblatt, N., Crenshaw, J. R., Grabiner, M. D. 2010 Variation in trunk kinematics influences variation in step width during treadmill walking by older and younger adults. *Gait Posture*. 31, 461-464.

van Leeuwen, A. M., van Dieen, J. H., Daffertshofer, A., Bruijn, S. M. 2020 Step width and frequency to modulate: Active foot placement control ensures stable gait. *bioRxiv*.

Reimann, H., Fetzrow, T., Jeka, J. J. 2018 Strategies for the control of balance during locomotion. *Kinesiology Review*. 7, 18-25.

Bauby, C. E., Kuo, A. D. 2000 Active control of lateral balance in human walking. *J Biomech*. 33, 1433-1440.

Kim, M., Collins, S. H. 2017 Once-per-step control of ankle push-off work improves balance in a three-dimensional simulation of bipedal walking. *IEEE Transactions on Robotics*. 33, 406-418.

2. Plots in Figure 3 could probably be increased in size to improve visibility.

We have increased the size of this figure.

Reviewer 3

Comments to the Author(s)

The authors used a device, in two variants, to provide elastic support to participants during walking. Participants wore a waist belt attached to a frame that was connected to lateral supports by springs mounted on horizontal sliders. Two experiments tested two different frames that differed in allowing or constraining rotation in the frontal plane. Both frames allowed to either lock or release a constraint on rotation in the transverse plane. The main goal was to investigate the effects of lateral stabilization on mechanical and metabolic gait features, and whether the constraint in the transverse plane makes a difference. The results do show an effect of stabilization on some gait parameters, but not others that other studies found affected, most notably energy cost. Releasing the transverse constraint increased the rotation in that direction, but not up to the level during normal walking.

This study is one incremental step in a larger body of work investigating lateral mechanical stabilization during walking. One problem with lateral stabilization is to design a device that constrains some degrees of freedom in the desired fashion, while leaving others unconstrained. This study makes some interesting contributions to this field, but suffers from presenting these

in a way that I found somewhat hard to follow, which might be a symptom of an underlying lack of clarity in the experimental design.

We wish to thank the reviewer for their comments, which we feel really helped improve the manuscript. Below, we provide a point-by-point reply, and any changes made in the manuscript have been marked. We would like to mention explicitly here that we really value the fact that the reviewer took the time to look at our code and raw data, and helped us discover a mistake, which of course should've been corrected before submission. Based on this, we have reanalyzed our data, which has led to slightly different results than the previous version of the manuscript.

Major Comments:

1. The main goal of this study is to compare the effect of lateral stabilization with and without a constraint on pelvis rotations in the transverse plane, implemented by locking or releasing lateral sliders that allow the connection point between the pelvis and the stabilization frame on either side to move in the anterior-posterior direction, labelled “free” and “restricted” walking. This is compared with a third condition of “normal” walking, without any stabilization. The authors performed two experiments, where Experiment 2 differed from Experiment 1 by (a) adding a two-minute familiarization period for each condition and (b) using a mechanically simpler stabilization device that removed the outer frame, thus making the device lighter, but also constraining pelvis rotation in the frontal plane. I don't really understand the rationale for two different experiments here. The authors state that “Experiment 2 supplemented Experiment 1, as it considered several potential confounding factors in the design and set-up of Experiment 1” — but these confounding factors are not listed or discussed in detail. The main difference is the removal of the outer frame in the device, thus constraining pelvis rotations in the frontal plane. This seems to *add* a potential confounding factor, rather than removing it. I can see how the absence of a familiarization period in Experiment 1 is a potential confounding factor, but on its own, I wouldn't say that provides sufficient rationale for a whole new experiment. Could you please elaborate in what manner the addition of the frontal plane constraint removes a confounding factor, and what other confounding factors that Experiment 2 avoided? 
We agree with Reviewer's comment that the rationales for performing Experiment 2 were not clear in the submitted version of our manuscript. The lack of rationales were also considered in the first major comment of Reviewer 1. As stated in the response to the first comment of Reviewer 1, we failed to reach a significant reduction of energy cost in stabilized conditions in Experiment 1 which was in contrast to the results reported by some of previous studies. The weight of the frame used in Experiment 1 was considered as a potential confounding factor which could offset some of the energy cost savings of external lateral stabilization. Removing the outer frame decreased the weight of frame (i.e. 3.0 kg lighter) and made it more similar to the frame used by previous studies. We have explained this issue as follows:

Page 5, lines 15-22 and page 6, line 1:

“Experiment 1 was performed to test the effect of external lateral stabilization with and without constrained transverse pelvic rotation on mechanical and metabolic gait features (Figure 1. & Figure 2. A). However, the results of Experiment 1 failed to reach the significant reduction of energy cost in the stabilized condition which was reported by some previous studies [1-3]. The potential effects of the frame weight on energy cost was considered as a potential confounding factor of Experiment 1. Additionally, the lack of habituation time to allow the participants for full familiarization with the set-up was considered as another reason for our inability to reduce energy cost in stabilized condition. Having the same aim and taking these potential confounding factors (weight of frame and habituation time) into account, we performed Experiment 2 to supplement Experiment 1.”

2. Another factor is that in the device used here, the connection between the body-worn frame and the external anchoring frame was not fixed, but mounted on trolleys that were free to move in the anterior-posterior direction. This, the authors argue, removes a back door for using the lateral springs to generate forward-pulling forces that decrease the metabolic cost of walking. This is a very interesting point, because this would imply that the results from other experiments showing that lateral stabilization reduced metabolic cost might be spurious results of the forces in the anterior-posterior direction implicitly generated by the mechanism. This point is only made in the discussion, maybe because the authors also only realized this after seeing their results? If this was something the authors were aware of before conducting the experiment, I would suggest adding this to the list of research questions they attempted to answer. Otherwise, I suggest that you clarify that this result was surprising and the explanation was generated post-hoc. In the latter case, I would also recommend bringing this up a bit later in the Discussion section: currently it is the first phenomenon that is discussed in-depth, after the general first paragraph, suggesting that this is the main result, which is at odds with it not being mentioned as a research question earlier.

The reviewer is correct that we only realized this after our experiment. We agree with the reviewer that it would be good to mention this somewhere later in the discussion, and clarify that it was nonexpected. We have shifted this discussion to the fourth paragraph, after the general first paragraph.

3. The authors made all data and analysis scripts available, which is great. Looking over what's in this cloud drive, however, I noticed some weird issues. One issue is that some of the variables have very large jumps, as seen in software/Plots, for example the Right Arm Swing in Subject 1, Trial 1 or Subject 14, Trial 5. This might be a problem with calculating angles from the rigid body orientation given by three markers on the cluster, since the jumps seem to be roughly around 90deg. Similar jumps occur in ML Pelvis Displacement, though, e.g. Subject 1, Trial 6; Subject 5, Trial 9; Subject 9, Trial 4. Another issue is gaps in the data, where some of the trajectories will

just disappear for some of the gait cycle, e.g. in Right Arm Swing in Subject 10, Trial 9 or Transverse Pelvis Rotation Subject 1, Trial 4, where around 60% of the gait cycle *all* data is missing, similarly for Subject 6, Trial 5 around 0-20%. I did not go through the analysis code in detail, so it is possible that these are just intermediate results, before such issues have been weeded out by the authors, although the readme.docx seems to suggest that this is not the case. If this is the case and these artifacts are still part of the data as analyzed in the manuscript, then I suggest that the authors go back to the data processing stage and take a close and careful look at where they come from and how to avoid them. In some cases, removing a small number of problematic gait cycles might be sufficient, but in other cases, all data seems to be missing for part of the gait cycle, and I don't know of a good way to deal with this.

We thank the reviewer for spotting these errors in our data analysis. This is one of the reasons why we also share the data (and code); to make sure that (due to some unforeseen circumstances) we don't end up publishing rubbish. So, we are really happy that you spotted this mistake. Indeed, some of the data was quite noisy, part of which was caused by malfunctioning of the equipment (there were renovations on the floor where the lab is located, and only after these, we discovered that dust on our Optotrak lenses may have caused us quite some problems). This has really been a head-breaker for us, and has left us with some data we were not able to salvage. For the current dataset however (after carefully inspecting all the graphs, which we intended to do in the first place, but forgot), we are confident in the results). We have adjusted the data-analysis, to prevent such issues from happening. In particular, we removed any parts in the data where large spikes in acceleration occurred. Furthermore, instead of taking the mean of three markers as our basis for translations, we now only use the marker with the least (or no) jumps. For the arm swing, we found that always at least one marker was very "jumpy", such that angle calculations became impossible. Hence, we used the translation of one of the markers with respect to the thorax marker instead, which provides another measure of arm swing. Of course, all of this has led to substantial changes throughout the manuscript. This has now been mentioned in the text:

Page 9, lines 1-2:

"Kinematic data from the Optotrak system were not filtered, but large jumps in the data were removed, and gaps of <10 samples were interpolated using a shape preserving spline algorithm."

Minor issues and questions

1.p.7, l.32 "the pelvis was restricted from rotating in the transverse plane" — what exactly was the mechanism here to release or lock this constraint? Figure 2 seems to imply that some slider can be locked. Please add some detail here.

We have added more details as follows:

Page 6, lines 16-21:

“To restrict/ allow transverse pelvis rotation, two horizontal sliders between waist belt and inner frame were used. Two screws were embedded on each horizontal slider. In one condition, the screws were fastened and the pelvis was restricted from rotating in the transverse plane. In another condition, the screws were loosened and participants could rotate their pelvis with minimal friction between the waist belt and horizontal sliders on the inner frame.”

2. p.7, l.45 “normal walking” — did normal walking consist of walking while wearing the device but without lateral springs attached, or of walking entirely without the device? Please add this information to the text. If the latter, did you confirm that the normal arm swing was not impeded by the frame at all? The hands can reach quite far ahead of the body in normal arm swing, especially at fast walking speeds, and from Figures 1 and 2 it seems that arm swing might be impeded.

We have added more information, indicating that subjects walked in the Normal condition without wearing the frame or being attaching to lateral stabilization set-up as follows:

Page 7, lines 4-5:

“Participants were measured in three conditions (normal walking, entirely without wearing the frame and without being attached to the lateral stabilization set-up (Normal),...”

About arm swing, we have added the following sentences, indicating that arm swing was not restricted by our frames:

Page 6, lines 15-16:

“The distance between the inner and outer frames allowed normal arm swing during walking and participants were able to swing their arms through the full range of motion.”

Page 7, lines 21-22:

“Two stiff ropes attached to the frame on either side, joined each other at 0.5 m from the frame, providing space for full range of motion that arms can swing.”

3. p.9, l.8 “Clusters of three infrared markers were attached to ... the left and right arms” — please specify where exactly on the arms the clusters were placed.

We have added more information here as follows:

Page 8, lines 9-11:

“Clusters of three infrared markers were attached to the thorax (over the T6 spinous process), the pelvis, the waist belt of the frame (see Figure 2 A. & B.), the left and right arms (over the lateral and middle part of the humerus segment) and the heels.”

4. p.9, l.29/50 Please specify which convention you used for axes of rotation in calculating Euler angles.

This has now been mentioned in the text:

Page 9, lines 7-9:

“Using Euler angles (zxy sequence), the time series of transverse and frontal pelvis rotations, and transverse thorax rotation were calculated from the segment orientation matrices.”

5. p.9, l.46 — how did you identify the heel strike events?

Heel strikes were identified as the minimum in the vertical position of the heel marker, and identified heelstrikes were visually inspected. This has now been mentioned in the text:

Page 9, lines 3-5:

“Heel strike events were identified as the minimum in the vertical position of the heel marker, and identified heel strike events were visually inspected”

6. p.10, l.10f: anterior-posterior pelvis displacement — at this point it seems odd that the ap-displacement is defined differently than the other displacements/angles. Is this because the ap-displacement is relevant in terms of the implicit forces in this direction that were *not* applied by your device due to the trolley connection? If so, it would help to explain that at this point, and maybe give this variable a different name to more clearly differentiate it from the other means.

Indeed, we calculated this variable in this way because it is relevant in terms of the forces that would be applied to a subject would we not have had the trolleys. The reviewer is correct that we could have stated this more explicit. As a matter of fact, the AP motion within a gait cycle may also be relevant, as it gives more of an indication if and how the gait pattern changes. Hence, we have now updated the manuscript to reflect these ideas more clearly. Specifically, we now define both the amplitude of anterior-posterior drift over a trial, and anterior-posterior pelvis displacement as the differences between maximum and minimum displacements per gait cycle and then median of amplitudes over gait cycles for each trial. In this version of our submission, we have included both measures. This has led to several changes in the manuscript, as outlined below:

Page 9, lines 9-12:

“The time series of angular (i.e. transverse and frontal pelvis rotations, and transverse thorax rotation) and displacement (i.e. medio-lateral, anterior-posterior, vertical pelvis displacements, and arm swing (i.e. anterior-posterior position of arm with respect to anterior-posterior position of thorax)) variables were time normalized to 0-100% for each gait cycle.”

Page 9, lines 14-17:

“Moreover, to explore whether lateral stabilization constraints the anterior-posterior drift of participants over a trail, we calculated the amplitude of anterior-posterior drift as differences between maximum and minimum of anterior-posterior pelvis displacements over a walking trial.”

7. p.10, l.35 “distances between ... foot placements” — what exactly was the location of the foot placement here?

We don't quite understand the reviewers question here, but upon re-reading, also see that our wording of that sentence was a bit odd. We have updated that sentence, it now reads:

Page 10, lines 1-3:

“Step length and step width were defined as the median of the distances between both feet in anterior-posterior and medio-lateral foot directions at heel strike, respectively.”

We hope this resolves your issue.

8. p.10, l.37 “for the step length, we calculated the average over legs, since nonsignificant differences were found between left and right step lengths” — what about step width? If it was not significant, did you also pool data? If it was significant, did you use it as a factor in the statistical analysis?

Step width can, per definition, not differ between left and right steps, as it would entail sideways walking in some way. Hence, we indeed averaged over legs. We now mention this in the manuscript.

Page 10, lines 4-5:

“Step width was calculated likewise as the average over left and right steps.”

9. Most figures: The legend indicates that the whiskers of the box-and-whisker plots cover the whole range of the data, from “Min” up to “Max”. However, many of the individual data points are outside of this range, so that cannot be correct. Please clarify. Also, it would help to add some

horizontal jitter to the individual data points, so they are still distinguishable when multiple data points are close together.

The reviewer is correct, this was not correct. We have fixed the boxplots, covering the whole range of data from Min up to Max. We have added some horizontal jitter to the individual data point and they are distinguishable.

Language

10. p.6, l.1 constrains → constraints

11. p.9, l.55 matrixes → matrices

12. multiple locations: capitalize “Experiment 1”

We have changed all of these as suggested.

1. Dean, J.C., N.B. Alexander, and A.D. Kuo, *The effect of lateral stabilization on walking in young and old adults*. IEEE Transactions on Biomedical Engineering, 2007. **54**(11): p. 1919-1926.
2. Donelan, J.M., D.W. Shipman, R. Kram, and A.D. Kuo, *Mechanical and metabolic requirements for active lateral stabilization in human walking*. Journal of biomechanics, 2004. **37**(6): p. 827-835.
3. Ortega, J.D., L.A. Fehلمان, and C.T. Farley, *Effects of aging and arm swing on the metabolic cost of stability in human walking*. Journal of biomechanics, 2008. **41**(16): p. 3303-3308.
4. Ijmker, T., S. Noten, C. Lamothe, P. Beek, L. van der Woude, and H. Houdijk, *Can external lateral stabilization reduce the energy cost of walking in persons with a lower limb amputation?* Gait & posture, 2014. **40**(4): p. 616-621.
5. Ijmker, T., H. Houdijk, C.J. Lamothe, P.J. Beek, and L.H. van der Woude, *Energy cost of balance control during walking decreases with external stabilizer stiffness independent of walking speed*. Journal of biomechanics, 2013. **46**(13): p. 2109-2114.
6. Kerrigan, D.C., P.O. Riley, J.L. Lelas, and U. Della Croce, *Quantification of pelvic rotation as a determinant of gait*. Archives of physical medicine and rehabilitation, 2001. **82**(2): p. 217-220.
7. Della Croce, U., P.O. Riley, J.L. Lelas, and D.C. Kerrigan, *A refined view of the determinants of gait*. Gait & posture, 2001. **14**(2): p. 79-84.
8. de CM Saunders, J., V. Inman, and H. Eberhart, *The major determinants in normal and pathological gait*. J Bone Joint Surg, 1953. **35**: p. 543-558.
9. Bruijn, S., O. Meijer, P. Beek, and J. Van Dieën, *Assessing the stability of human locomotion: a review of current measures*. Journal of The Royal Society Interface, 2013. **10**(83): p. 20120999.
10. Bruijn, S.M. and J.H. van Dieën, *Control of human gait stability through foot placement*. Journal of The Royal Society Interface, 2018. **15**(143): p. 20170816.
11. Liang, B.W., W.H. Wu, O.G. Meijer, J.H. Lin, G.R. Lv, X.C. Lin, M.R. Prins, H. Hu, J.H. van Dieën, and S.M. Bruijn, *Pelvic step: the contribution of horizontal pelvis rotation to step length in young healthy adults walking on a treadmill*. Gait & posture, 2014. **39**(1): p. 105-110.

Appendix B

Reviewer: 1

Comments to the Author(s)

This manuscript demonstrated that the typical methods used to provide lateral stabilization during gait provides additional effects aside from improving mediolateral stability. To remedy this issue, the authors presented two novel designs that aim to allow for frontal and transverse pelvis rotation that were restricted in previous studies. The first design had two additional degrees of freedom to allow for frontal and transverse pelvis rotation, while the second removed the frame that allowed for frontal pelvis rotation to reduce the weight of the device. The first experiment showed that no change in frontal pelvis rotation but also no change between free and restricted conditions for transverse pelvis rotation. The second experiment found reduced frontal pelvis rotation and significantly greater transverse pelvis rotation for free over the restricted condition, although it was still lower than normal walking.

Overall, this paper does an excellent job of describing the effect of mediolateral stabilization and the two novel designs presented offered intuitive solutions for observed issues. Based on the provided author responses to reviewers, the authors have thoroughly incorporated feedback from the previous review, and the manuscript has greatly improved. In my opinion, I believe that this manuscript should be accepted, but I do have a few comments that I believe would enhance the paper.

We wish to thank the reviewer for the comments, which we feel really helped improve the manuscript. Below, we provide a point-by-point reply, and any changes made in the manuscript have been marked.

Major Comments:

1. Page 14 Line 15: The "normal" frontal pelvis rotations refers to no significant difference among Normal, Free, and Restricted for Experiment 1, but the "more normal" transverse pelvis rotation for Experiment 2 is because Free is significantly greater than Restricted. One might misinterpret the latter as no significant difference among the three conditions, although the wording "more normal" helps. In addition, it is ambiguous whether "the provided rotational degrees of freedom" refers to Free or Restricted or both. I suggest an amendment to help clarify the sentence: "... allowing free transverse pelvis rotation in our new set-ups resulted in normal frontal pelvis rotation (Experiment 1), or more normal (i.e. greater) transverse plane pelvis rotation (Experiment 2)..."

If we revise this sentence based on the suggested amendment, one might interpret that the normal frontal pelvis rotation in Experiment 1 was due to free transverse pelvis rotation. However, this was not the case. The normal frontal pelvis rotation in Experiment 1 was due to the joint between inner and outer frames (see number 6 in Figure 2.) which provided a degree of freedom. Therefore, we have revised this sentence as follows:

Page 14, lines 15-17:

"...,our set-up that allowed free frontal and transverse pelvis rotations (Experiment 1) resulted in normal frontal plane pelvis rotation. While our set-up that only allowed free

transverse pelvis rotation (Experiment 2) resulted in more normal (i.e. greater) transverse plane pelvis rotation.”

2. Page 14, Line 18: I think it would be helpful to be more specific which experiment is being referred to in this line as here was no observed reduction in the frontal pelvis rotation in Experiment 1 (per figure 3). Extending from this point, I would encourage the authors to comment on why there was no significant difference in the frontal pelvis rotation for Experiment 1 in the restricted condition. Based on the information provided in the introduction, I would have expected there to be a reduction, especially in the restricted condition.

We have referred to the experiments as follows:

Page 14, lines 19-20:

“The reduced frontal and transverse pelvic rotations might confound the interpretation of previously reported results. For instance, our new set-ups resulted in normal frontal (Experiment 1) and more transverse pelvis (Experiment 2) rotations.”

We have just manipulated the transverse pelvis rotation in Experiment 1 and the joint between inner and outer frames (see number 6 in Figure 2.) provided a free degree of motion in frontal plane in this experiment. Thus, participants could freely move their pelvis in frontal plane in all stabilized conditions (i.e. Free and Restricted). To clarify this, we have added more details in the experimental protocol of experiment 1 as follows:

Page 7, lines 9-11:

“Participants were able to freely move their pelvis in frontal plane because of the joint between inner and outer frames. Thus, the frontal pelvis rotation was not restricted in both stabilized conditions (i.e. Free and Restricted).”

We have also added more details in the discussion:

Discussion, page 14, lines 10-14:

“The frame used in Experiment 1 included an inner and an outer frames, which were attached to each other and provided a free rotational degree of motion in the frontal plane between pelvis and frame in all stabilized conditions, however the frame used in Experiment 2 did not have an outer frame and it was more similar to the frame used by previous studies [1, 2].”

3. Page 14, Line 23: In this section, the authors suggest that the reduction in the vertical center of mass displacement could be responsible for the reduction in metabolic cost. Although valid references are given to support this statement, the effect of vertical center of mass displacement still quite disputed throughout the literature (see references below). To be consistent with past literature, it would be helpful include that reduced vertical displacement could be one cause of reduced metabolic cost, but since there is no consensus on the effect of the reduced center of mass displacement, there may be additional causes that reduce metabolic cost.

References:

Keith E. Gordon, Daniel P. Ferris, Arthur D. Kuo; Metabolic and Mechanical Energy Costs of Reducing Vertical Center of Mass Movement During Gait; Archives of Physical Medicine and Rehabilitation; Volume 90, Issue 1; 2009; Pages 136-144,

Justus D. Ortega and Claire T. Farley; Minimizing center of mass vertical movement increases metabolic cost in walking; Journal of Applied Physiology 2005 99:6, 2099-2107

To take this inconsistency in the literature into account, we have added following sentences:

Page 14, lines 19-23 and page 15, lines 1-10:

“The reduced frontal and transverse pelvic rotations might confound the interpretation of previously reported results. For instance, our new set-ups resulted in normal frontal (Experiment 1) and more transverse pelvis (Experiment 2) rotations. The reduced frontal and transverse pelvis rotations in previous and partly in our set-up, need to be considered as the results of the physical constraints of apparatus and cannot be attributed to a strategy to control gait stability. Moreover, it has been reported that transverse pelvis rotation, as one of the gait determinants [3], has some mechanical and metabolic benefits as it reduces vertical center of mass displacement [4, 3] and increases step length [4]. There is no clear consensus, however, on whether reduced vertical center of mass displacement induces a metabolic cost reduction [4, 3] or not [5, 6]. Moreover, the effect of transverse pelvis rotation on step length is still disputed throughout the literature [4, 7]. Assuming that transverse pelvis rotation reduces vertical center of mass displacement and metabolic cost and that transverse pelvis rotation increases step length, restricted transverse pelvis rotation induced by lateral stabilization might offset these benefits. In line with this, the restricted step length in our Experiment 1 could be due to the restricted transverse pelvis rotation and unaffected step length in our Experiment 2 could be considered as the mechanical benefits of increased transverse pelvis rotation.”

Minor Comments:

4. Page 4 Line 19: Please include if transverse pelvis rotation increases or decreases step length.

Please read our response to the comment 3.

1. Ijmker, T., H. Houdijk, C.J. Lamoth, P.J. Beek, and L.H. van der Woude, *Energy cost of balance control during walking decreases with external stabilizer stiffness independent of walking speed.* Journal of biomechanics, 2013. **46**(13): p. 2109-2114.
2. Ijmker, T., S. Noten, C. Lamoth, P. Beek, L. van der Woude, and H. Houdijk, *Can external lateral stabilization reduce the energy cost of walking in persons with a lower limb amputation?* Gait & posture, 2014. **40**(4): p. 616-621.
3. Kerrigan, D.C., P.O. Riley, J.L. Lelas, and U. Della Croce, *Quantification of pelvic rotation as a determinant of gait.* Archives of physical medicine and rehabilitation, 2001. **82**(2): p. 217-220.

4. de CM Saunders, J., V. Inman, and H. Eberhart, *The major determinants in normal and pathological gait*. J Bone Joint Surg, 1953. **35**: p. 543-558.
5. Gordon, K.E., D.P. Ferris, and A.D. Kuo, *Metabolic and mechanical energy costs of reducing vertical center of mass movement during gait*. Archives of physical medicine and rehabilitation, 2009. **90**(1): p. 136-144.
6. Ortega, J.D. and C.T. Farley, *Minimizing center of mass vertical movement increases metabolic cost in walking*. Journal of Applied Physiology, 2005. **99**(6): p. 2099-2107.
7. Liang, B.W., W.H. Wu, O.G. Meijer, J.H. Lin, G.R. Lv, X.C. Lin, M.R. Prins, H. Hu, J.H. van Dieën, and S.M. Bruijn, *Pelvic step: the contribution of horizontal pelvis rotation to step length in young healthy adults walking on a treadmill*. Gait & posture, 2014. **39**(1): p. 105-110.